# Water as a gas separation membrane

Kian P. Lopez[1], Max Saffer-Meng [1], Mohammad Allouzi[2,3], Yukai Tomsovic[2,4], Joshua N. Sherrit[1], Sasha R. Neefe[1], Patrick O. Saboe[5], Mou Paul[5], Abhishek Roy[5] & Anthony P. Straub [2,3,4] ✉

Efficient gas separation membranes are essential for carbon capture, biogas upgrading, and hydrogen purification. Inspired by how plants absorb $CO_2$ through water, we present a membrane platform that uses liquid water as the selective layer. Hydrophilic sub-100-nm pores stabilize water through strong capillary forces, enabling operation at feed pressures above 72 bar under dry and humid conditions. Selectivity is governed by gas solubility in water, while permeance is tuned by adjusting the water layer thickness. Reducing this thickness below 200 nm yields $CO_2$ permeances up to 11,600 gas permeation units with $CO_2$:$N_2$, $CO_2$:$CH_4$, and $CO_2$:$H_2$ selectivities of 40, 26, and 31, respectively, surpassing the performance of state-of-the-art membranes. Operation is sustained for over a week without water loss, and performance scales using commercially available porous polymer supports under mixed-gas crossflow conditions. Water's dissolution-based transport avoids saturation and reaction-rate limits, enabling a robust, high-performance, and environmentally benign gas separation platform.

Gas separations are crucial to industrial and environmental processes such as carbon capture, biomethane purification, and syngas upgrading, where efficient separation of $CO_2$ from gases like $N_2$, $CH_4$, and $H_2$ is necessary[1–3]. Membrane-based gas separations have the potential to be up to ten-times more energy efficient than conventional technologies, such as amine scrubbing and cryogenic separation, while also offering simpler operation and modular design[4–8]. However, current membranes are fundamentally limited by the trade-off between permeance and selectivity, where most membranes exhibit moderate selectivity, but their throughput is constrained by low $CO_2$ permeance[9,10]. Furthermore, many widely considered membrane materials are susceptible to plasticization or physical degradation when exposed to high-pressure and humid conditions, limiting their use in real-world applications[11–16].

Supported liquid membranes, which consist of a porous support impregnated with a gas-selective liquid phase, have shown promise in addressing some limitations of current materials. The liquid used, commonly an ionic liquid or amine, can be tailored to selectively interact with $CO_2$ via favorable physical and chemical interactions,

allowing for exceptionally high $CO_2$ selectivities. However, the gas permeance of supported liquid membranes is often limited by low gas diffusivity, difficulty in making thin liquid layers[15,17], and slow reaction kinetics in certain liquids that rely on chemical reactions with $CO_2$[18]. In addition, many supported liquid membranes exhibit performance losses as $CO_2$ partial pressure increases due to saturation of chemical sorption sites, and experience liquid displacement or blowout at relatively low pressures (less than 5 bar)[19,20]. Thus, there is a need to explore material systems that are more permeable, selective, and robust under pressure.

In nature, water plays a central role in gas separations: the two largest carbon sinks on Earth, oceans and plants, both rely on liquid water as a medium to absorb $CO_2$[21,22]. Plants, in particular, have evolved to uptake $CO_2$ by dissolving it in water-filled nanochannels lining the walls of leaf mesophyll cells (Fig. 1a)[23–25]. The gas–liquid interfaces in these channels both serve as a platform to absorb $CO_2$ for photosynthesis and sustain the large negative pressures (up to 150 bar) needed to drive water up from the roots using strong capillary forces[26,27]. Nature therefore leverages two unique properties of water

[1]Department of Chemical & Biological Engineering, University of Colorado Boulder, Boulder, CO, USA. [2]Department of Mechanical and Process Engineering, ETH Zürich, Zürich, Switzerland. [3]Department of Civil, Environmental & Architectural Engineering, University of Colorado Boulder, Boulder, CO, USA. [4]Materials Science & Engineering Program, University of Colorado Boulder, Boulder, CO, USA. [5]National Laboratory of the Rockies, Golden, CO, USA. ✉e-mail: astraub@ethz.ch

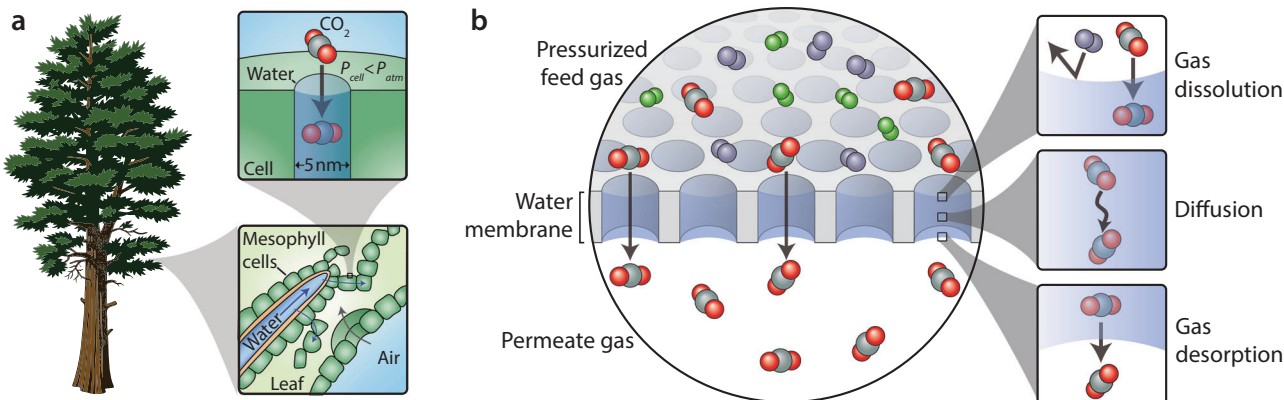

**Fig. 1 | Water membranes for gas separations. a** Schematic diagram of $CO_2$ uptake in trees, where $CO_2$ dissolves into water-filled nanochannels in the wall of leaf mesophyll cells. Capillary forces and water evaporation generate lower pressures inside the cell water channels, $P_{cell}$, than in the atmosphere, $P_{atm}$. These negative pressures, up to 150 bar, drive water uptake through the roots and water flow up the xylem water channels. **b** Schematic diagram of water-based membranes for gas separations, where a layer of water is stabilized within hydrophilic pores. To transport through the membrane, feed gases must dissolve in the water, diffuse through the layer in the aqueous phase, and desorb on the permeate side of the membrane. $CO_2$ is preferentially transported because of its higher solubility in water compared to other gases.

in its $CO_2$ uptake mechanism: (i) water has high $CO_2$ solubility through physical dissolution and (ii) water has a high surface tension that allows it to remain stable in nanoscale capillaries under large pressure differences[25]. Despite these promising properties of water for gas separations, relatively few studies have explored water in an engineered gas separation membrane. Prior work has incorporated carbonic anhydrase, ionic liquids, or other reactive chemicals into an aqueous layer to enable facilitated transport[3,17,28], but membranes that mimic the high-pressure tolerance of capillary-stabilized water observed in leaves to improve $CO_2$ separations remain unexplored.

Inspired by the mechanisms of carbon capture in nature, we explore membranes that use water as a selective layer for gas separations. These membranes employ hydrophilic nanopores to stabilize a thin, pressure-stable water layer (Fig. 1b). To transport through the membrane, gases must dissolve in the water, diffuse through the liquid layer, and desorb on the permeate side of the membrane. We use experiments and theory to probe gas transport behavior in these membranes, finding that selectivity is primarily governed by gas solubility in water, which results in high $CO_2$ permeance relative to other gases. By decreasing the thickness of the water layer, we show that gas permeance can be increased without sacrificing selectivity, allowing the membrane to exceed the perm-selectivity limits of state-of-the-art materials in key separations. Furthermore, we investigate the ability of nanoscale pores to stabilize the water layer under high applied pressure and enable days-long operation of the process without changes in performance. We also explore scale-up with commercial, large-scale porous polymer membranes and demonstrate consistent performance under mixed-gas conditions relevant to practical applications. Overall, this work provides a proof-of-concept demonstration for high-performance gas separation membranes leveraging liquid water.

## Results

### Principles of liquid water membranes

Gas transport through liquid water membranes proceeds through a solution-diffusion mechanism, in which gases first dissolve at the feed-side gas–liquid interface, diffuse through the water layer, and then desorb into the gas phase on the permeate side[29]. The dissolved gas concentration at the gas–liquid interface is defined by Henry's law:

$$C_g = K_H \cdot P_g \qquad (1)$$

where $C_g$ is the dissolved gas concentration, $K_H$ is the Henry's constant, and $P_g$ is the gas partial pressure. A difference in the gas partial pressure across the membrane, $\Delta P_g$, leads to a concentration gradient of dissolved gas across the membrane that drives gas flux, $J$, as described by Fick's first law of diffusion:

$$J = -D \frac{dC_g}{dx} \qquad (2)$$

where $D$ is the diffusion coefficient of the respective gas in water and $\frac{dC_g}{dx}$ is the concentration gradient of dissolved gas over length $x$. Combining Eqs. (1) and (2) yields a flux expression in terms of known parameters:

$$J = \frac{DK_H}{l} \Delta P_g \qquad (3)$$

where $l$ is the effective water layer thickness. Together, these relations demonstrate that transport is dictated by gas solubility, diffusivity in water, partial pressure gradients, and membrane thickness.

The solution-diffusion model reveals important features of water membrane behavior and design. Since gas diffusivities in liquid water are relatively similar across species, selectivity arises mainly from differences in solubility; these solubility differences are captured by the gas's Henry's constant, which describes the equilibrium partitioning of gas molecules into the water layer. Notably, $CO_2$ exhibits approximately 40 times higher solubility in water than other gases such as nitrogen, hydrogen, and methane, leading to high selectivity for $CO_2$. Moreover, since selectivity primarily depends on the Henry's constant, decreasing the thickness of the water layer can increase gas flux without sacrificing selectivity.

To operate effectively, the liquid water layer must remain stably confined within the membrane pores under an applied transmembrane pressure. The maximum pressure the membrane can withstand before gas displaces the water layer, $P_{max}$, is governed by capillary forces and described by the Young–Laplace equation:

$$P_{max} = \frac{2\gamma \cos\theta}{r} \qquad (4)$$

where $\gamma$ is the surface tension of the liquid-gas interface, $\theta$ is the intrinsic water contact angle on the membrane surface which serves as

an approximation for the receding contact angle, and $r$ is the pore radius.

In addition to pressure-driven displacement, the stability of water within the pores is also influenced by ambient humidity. At low relative humidities, the confined water phase will evaporate if liquid–vapor equilibrium is not sustained. Capillary forces of water in the nanopore inhibit evaporation by decreasing the equilibrium partial vapor pressure of water, as described by the Kelvin equation:

$$P_v = P_{v,0} \exp\left(-\frac{2\gamma V_w}{r_c RT}\right) \qquad (5)$$

where $P_v$ is the vapor pressure at the gas-water interface, $P_{v,O}$ is the equilibrium vapor pressure as a function of temperature, $V_w$ is the water molar volume in the pore, $R$ is the universal gas constant, $r_c$ is the radius of curvature of the liquid-vapor interface, and $T$ is absolute temperature. As ambient humidity decreases, the radius of curvature at the interface must also decrease for vapor–liquid equilibrium to be maintained. The minimum radius of curvature possible in a capillary is a function of the pore radius, $r$, and the intrinsic water contact angle, $\theta$:

$$r_{c,\,min} = \frac{r}{\cos\theta} \qquad (6)$$

If the ambient water vapor pressure drops below the equilibrium pressure corresponding to this minimum curvature, capillary drying occurs, and the water is removed from the pore. Thus, maintaining stable operation of liquid water membranes requires both pressure resistance and stability under low relative humidity, both of which depend on small, hydrophilic pores to retain the water layer within the membrane.

## Experimental gas transport behavior and membrane robustness

To experimentally demonstrate water membranes and validate theory, we tested gas transport performance of water-filled porous membranes using a pressurized gas feed (Fig. 2a). Our initial testing used custom-fabricated anodic aluminum oxide membranes with controlled pore diameters (40, 80, 120, 160, and 200 nm) that were impregnated with water simply by pipetting a small droplet on the surface. The thickness of the water layer was varied from 100 μm to 190 nm either by using membranes of different thicknesses (Fig. 2a, top inset) or by selectively functionalizing the surface to have a hydrophilic layer of controlled thickness (Fig. 2a, bottom inset; Supplementary Note 1; Supplementary Figs. 1–3; and Supplementary Table 1). The aluminum oxide membranes provided an ideal platform to explore the fundamental behavior and performance limits of water membranes because (i) they have uniform isopores of known pore size and porosity (Fig. 2b, c and Supplementary Fig. 4), (ii) they experience minimal deformation under pressure, allowing for reliable measurements, and (iii) they can be functionalized to trap water on the top surface using available surface science techniques (Fig. 2d). Although aluminum oxide membranes were advantageous for proof-of-concept testing and model validation, lower cost and more readily scalable materials will likely be used in practice, as we demonstrate later. Characterization of the fabricated membranes is detailed in Supplementary Note 2 and Supplementary Figs. 4–10.

Experimental evaluation of the liquid water membranes confirmed their ability to resist water displacement under high applied gas pressures. Young–Laplace theory (Eq. (4)) predicts that displacement pressure scales inversely with pore diameter, and experimentally measured displacement pressures of membranes with different pore diameters from 40 to 200 nm demonstrated quantitative agreement with values modeled using Young–Laplace (Fig. 2e). The highest measured water displacement pressure was 72 bar in the membrane

with 40-nm diameter pores. This pressure exceeds the requirements for most applications, including natural gas sweetening and syngas upgrading, where the feed gas pressure can be 20–70 bar[30,31]. The maximum measured displacement pressure is also at least three times higher than values for supported ionic liquid membranes in the literature, which range from 0.1–20 bar[20,32]. We note that it is possible to achieve even higher displacement pressures using membranes with sub-40-nm pore diameters, but safety limitations in our experimental setup prevented testing beyond 72 bar.

Gas flux measurements through liquid-water membranes agree with predictions based on gas dissolution and diffusion theory. The measured gas flux increased linearly with feed gas pressure for $CO_2$ and $N_2$ pure gases (Fig. 2f). The $CO_2$ flux was approximately 40-times greater than the $N_2$ flux, consistent with the difference in aqueous solubility between the two gases. Gas fluxes modeled using solution-diffusion theory with Henry's law for gas solubility and Fick's first law for diffusion (Eqs. 1–3) showed strong agreement with experimental values using zero fitting parameters.

Permeation experiments with different gases demonstrated that selectivity in the water membrane is primarily governed by the aqueous solubility of each gas (Fig. 2g). Experimentally measured gas permeances increased with aqueous gas solubility for $N_2$, $H_2$, $CH_4$, $O_2$, and $CO_2$ (Supplementary Table 2). Since most gases have similar diffusion coefficients in water, differences in experimental gas permeances were well-described by the Henry's law constant (Supplementary Table 3 and Supplementary Fig. 11). $CO_2$ exhibited a 24–40 times higher permeance (average of 11,600 gas permeation units, GPU; 1 GPU = $10^{-6}$ cm³ [STP] cm⁻² s⁻¹ cmHg⁻¹) than other gases due to its uniquely high solubility arising from strong chemical interactions with water and polarizability.

Decreasing the water layer thickness led to a proportional increase in gas permeance without altering selectivity (Fig. 2h). To probe the impact of water layer thickness on transport rates, we fabricated membranes with water layer thicknesses spanning three orders-of-magnitude: 20, 50, and 100 μm thick alumina membranes were tested with water filling their entire thicknesses (Supplementary Fig. 12) in addition to membranes with 190 nm, 457 nm, and 1.7 μm thick water layers fabricated using controlled surface modification (Supplementary Notes 1 and 2). Since transport resistances in the membrane are dominated by aqueous-phase diffusion in the liquid water layer, the observed $CO_2$ permeance of the water layer was inversely proportional to the membrane thickness, increasing from 46 to 11,600 GPU as the thickness decreased from 50 μm to 190 nm. In contrast, prior supported liquid membranes typically employ much thicker active layers on the order of tens of microns, constraining their achievable $CO_2$ permeances to below 1000 GPU[17,33]. The selectivity between CO2 and N2, which is primarily dependent on gas solubility, was maintained between 31 and 40 for all membranes regardless of their thickness, showing that permeance can be increased by nearly three orders of magnitude by decreasing thickness without compromising selectivity.

Long-term testing showed that the water membranes remained stable over at least eight days of continuous operation with a dry gas feed (Fig. 2i). Continuous gas permeation tests were conducted using feed gases with a relative humidity below 1% and an applied pressure of 6.89 bar (Supplementary Figs. 13 and 14). The permeate side of the membrane was exposed to water vapor from a condensed water droplet. During the eight-day test, $CO_2$ permeance showed low variation, and $CO_2$:$N_2$ selectivity remained consistent between the first and eighth day of testing. This stability with dry feed conditions can be explained by the vapor pressure depression of water in nanoscale capillaries described by Eq. (5), with corresponding values given in Supplementary Table 4. By reducing the equilibrium vapor pressure of water in the pores, nanoscale confinement of water both increases the humidity threshold for evaporation and decreases the driving force for

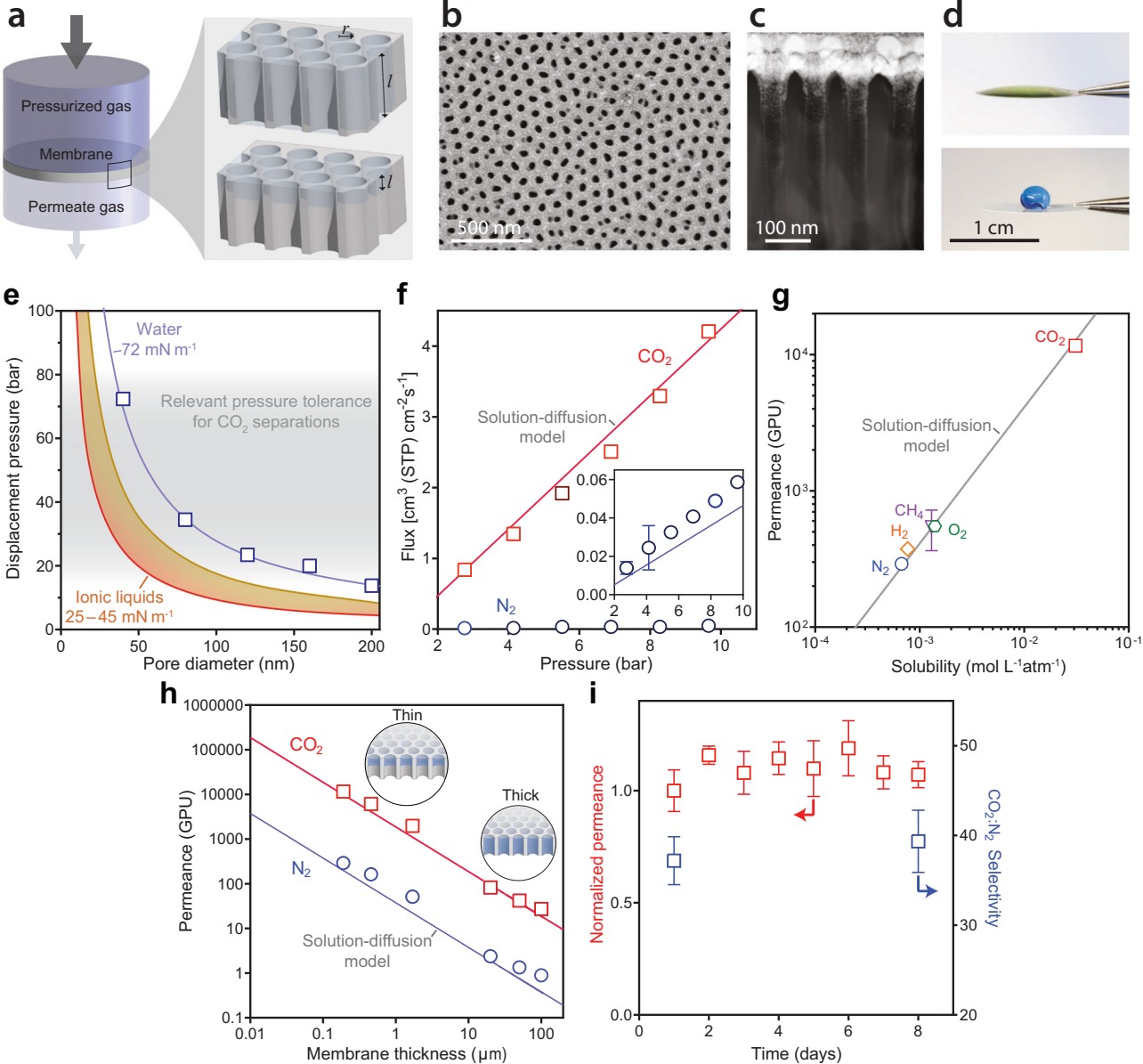

**Fig. 2 | Gas separation performance and robustness of the water membrane.**
**a** Schematic of testing setup for gas transport measurements. Inset shows membranes where water fills the entire pore thickness (top) and membranes selectively coated with a hydrophilic water-trapping layer in the upper pore (bottom). In both cases, the pore radius, $r$, and water layer thickness, $l$, could be controlled.
**b** Scanning electron micrograph of the top surface of the anodic alumina membrane with a 40-nm pore diameter. **c** Scanning transmission electron microscopy high-angle annular dark-field image showing a cross section of the upper pore.
**d** Contact angle of water on the hydrophilic top surface of the membrane (top) and on the hydrophobic bottom surface of the membrane (bottom). **e** Measured displacement pressure of the water layer as a function of pore diameter for fabricated membranes (symbols). Theoretical displacement pressures obtained from the Young–Laplace equation using the surface tension of water (72 mN/m) and ionic liquids (25 and 45 mN/m) are also shown[43]. **f** Measured gas flux as a function of the

feed gas pressure for fabricated membranes with 40-nm pore diameters and estimated water layer thicknesses of 287 nm. Lines show gas flux predicted using the solution-diffusion model. Gas flux was normalized to the membrane porosity of 12% so that reported values are representative of the water-filled active pore area.
**g** Measured gas permeance as a function of aqueous solubility for different gases using a membrane pore diameter of 40 nm and an estimated water layer thickness of 190 nm (symbols), and simulated permeance using the solution-diffusion model with a diffusion coefficient of $2.00 \times 10^{-5} \, cm^2 \cdot s^{-1}$ (line). **h** Permeance as a function of the water layer thickness for measurements (symbols) and simulations (lines). **i** Long-term stability of liquid water membranes over 8 days of operation. Gas permeance was normalized to the initial value. The relative humidity in the feed was kept below 1% by continuously venting the feed gas. Error bars denote the mean ±1 s.d. for measurements from at least three distinct membrane samples.

evaporation when it does occur. Under the tested conditions, these factors slow water loss from the dry feed side and promote net condensation on the permeate side, allowing the water layer to be sustained with only minimal moisture. The humidity needed to maintain the water layer is governed by the pore-size-dependent equilibrium vapor pressure described by the Kelvin equation (Supplementary Fig. 15).

## Permeance and selectivity performance in gas separations

Conventional gas-separation membranes are constrained by a trade-off: increases in gas throughput come at the expense of selectivity. In water-based membranes, a key advantage is the ability to reduce the thickness of the water layer without introducing defects, which are a major limitation in polymeric and supported liquid membranes. Since selectivity in the solution-diffusion framework depends solely on

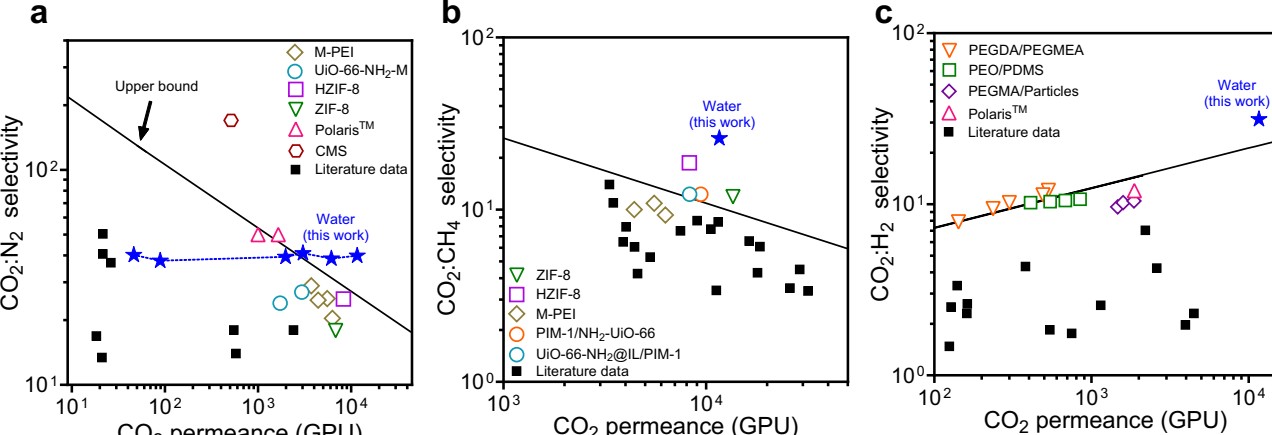

**Fig. 3 | Water membranes exceed current permeance and selectivity limits.** **a** Permeance-selectivity trade-off for $CO_2$:$N_2$ with the water membranes studied in this work (blue stars) compared to common values found in the literature (black squares) and the following high-performance membranes: Polaris[44], UiO-66-$NH_2$-M[45], ZIF-8[46], HZIF-8[5], CMS[47], and M-PEI[36]. The Robeson upper bound[9] assuming a thickness of 1 μm, is also shown. **b** Permeance-selectivity trade-off for $CO_2$:$CH_4$ compared to literature values and high performance membranes: HZIF-8[5], PIM-1/$NH_2$-UiO-66[48], ZIF-8[46], UiO-66-$NH_2$@IL/PIM-1[49], and M-PEI[36]. **c** Permeance-selectivity performance for $CO_2$:$H_2$ compared to the permeance-selectivity upper bound[50] and high performance membranes: PEO/PDMS[51], PEGMA/Particles[52], Polaris[53], and PEGDA/PEGMA[54]. Gas permeance is normalized to the active pore area of the membrane based on a porosity of 12%.

differences in solubility and diffusivity between species (Eq. (3)), decreasing the water layer thickness enhances gas permeance without affecting selectivity, enabling operation with both high throughput and suitable discrimination between gases.

Gas separation performance for $CO_2$:$N_2$ separations, which are critical in carbon capture, demonstrates the ability of liquid water membranes to increase permeance without sacrificing selectivity (Fig. 3a). Perm-selectivity trade-offs have historically limited membrane performance for $CO_2$:$N_2$ separations, where conventional membranes exhibit an inverse relationship between permeance and selectivity, captured by the Robeson upper bound[34-36]. In contrast, the liquid water membranes show increasing $CO_2$ permeance with decreasing water layer thickness, reaching values as high as 11,600 GPU with a water layer thickness of 190 nm while maintaining a consistent $CO_2$:$N_2$ selectivity of 40:1. This performance exceeds the Robeson upper bound and highlights the potential of liquid water membranes to outperform state-of-the-art materials reported in the literature (Supplementary Table 5). We note that any membrane governed by solution-diffusion should, in theory, be able to similarly increase permeance without sacrificing selectivity since selectivity arises from either solubility or diffusivity differences (Eq. (3)); in practice, however, fabricating conventional polymer and ionic liquid membranes that are thin and defect-free has been challenging, leading to the trade-off observed in Fig. 3a[10].

In addition to $CO_2$:$N_2$ separation, other gas separations such as $CO_2$:$CH_4$ and $CO_2$:$H_2$ are critical for applications including natural gas upgrading and syngas conditioning[37-39]. Such separations benefit from membranes that maintain suitable selectivity while also achieving high $CO_2$ permeance. Our membranes demonstrate selectivities of 26 for $CO_2$:$CH_4$ and 31 for $CO_2$:$H_2$, values that are competitive with state-of-the-art materials. More importantly, these selectivities are sustained at $CO_2$ permeances exceeding 11,000 GPU, surpassing the performance of conventional membranes that typically operate in the range of 100–5000 GPU (Fig. 3b, c). In addition, hydrogen sulfide ($H_2S$) and ammonia ($NH_3$), common contaminants in natural gas, biogas, and syngas, are predicted to permeate even faster than $CO_2$ through liquid water membranes, with theoretical selectivities of up to 3.4 and 1,800 for $H_2S$:$CO_2$ and $NH_3$:$CO_2$, respectively (Supplementary Fig. 16). Our results demonstrate that the high solubility-based selectivity and low resistance to transport provided by the sub-micron thick water

layer enables strong $CO_2$ separation performance across multiple gas pairs.

## Gas sorption capacity and kinetics
In addition to showing high $CO_2$ permeance, our experiments observed that the permeability of $CO_2$ in water membranes is maintained even at high $CO_2$ partial pressures up to 27 bar (Fig. 4a). This behavior contrasts with that of ionic liquids and some polymeric membranes, in which $CO_2$ permeability decreases at elevated partial pressures. The loss of $CO_2$ permeability in current materials limits $CO_2$ throughput and causes a loss of selectivity. Mechanistically, ionic liquids experience a loss in $CO_2$ permeability at high partial pressures because $CO_2$ sorption decreases as chemical sorption sites become saturated. Sorption of $CO_2$ in water relies on physical dissolution of gas molecules in water, rather than the formation of carbamate or other chemically bound species, and is weakly dependent on $CO_2$ partial pressure for all practical gas separation scenarios. As a result, $CO_2$ permeability in liquid water membranes is maintained even at elevated $CO_2$ partial pressures and, at $CO_2$ partial pressures above 1 bar, water surpasses the $CO_2$ permeability of many high-performing ionic liquids. This stability under high $CO_2$ loadings is particularly advantageous for treating flue gas or natural gas streams, which contain high concentrations of $CO_2$[40,41].

The physical dissolution mechanism of gases in liquid water membranes allows them to avoid the reaction-rate limitations encountered in membranes that rely on reaction-facilitated transport of a carrier species. Transport limitations in these systems can be described by the Damköhler number (Da) which quantifies the ratio of reactive transport to diffusive transport and can be calculated by

$$Da = \frac{k_r l^2}{D} \tag{7}$$

where $k_r$ is the rate constant for the gas interacting with the liquid ($s^{-1}$), $l$ is the liquid film thickness (m), and $D$ is the diffusivity of a gas through the liquid ($m^2$ $s^{-1}$). When the Damköhler number is much greater than unity, transport is diffusion limited. For a Damköhler number less than unity, the system is reaction rate limited, a regime that results in decreased selectivity for $CO_2$ over non-reactive species. In systems governed by amine- or carbonate-based chemistry, transport becomes

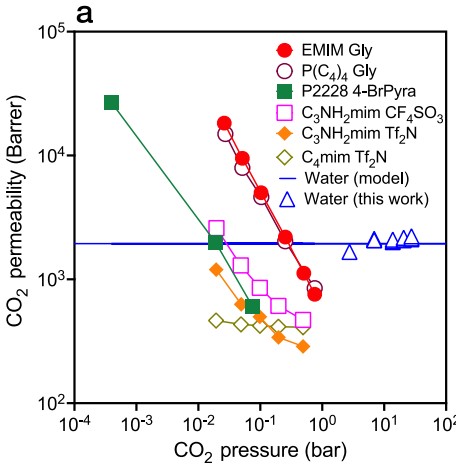

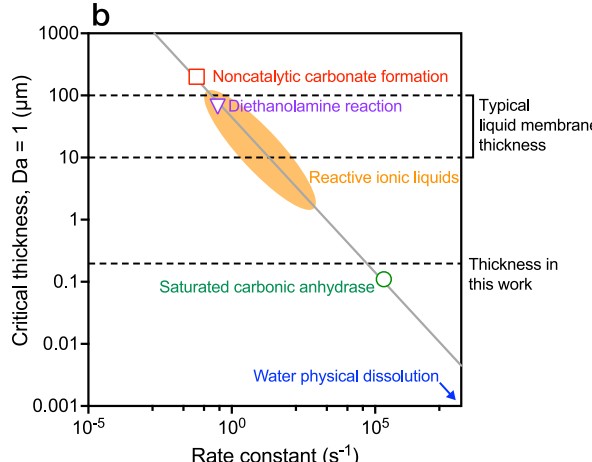

**Fig. 4 | Sorption capacity and dissolution kinetics of $CO_2$ in water. a** $CO_2$ permeability (Barrer) as a function of $CO_2$ partial pressure in supported ionic liquid and water membranes. At high $CO_2$ partial pressure, ionic liquids with high $CO_2$ solubility can become saturated, resulting in lower $CO_2$ permeability. Water maintains constant $CO_2$ permeability at high pressure because $CO_2$ transport is driven by physical dissolution[17,19,55]. **b** Critical thickness at which a system transitions from diffusion- to reaction-limited $CO_2$ transport (Da = 1) for different sorption and reaction processes: $CO_2$ reaction to form carbonate, $CO_2$ reaction with liquid amine, and limiting reaction rate of carbonic anhydrase enzyme[19,56–58]. Gas dissolution kinetics in water are rapid, and the equivalent critical thickness of water is below 1 nm. Typical thickness ranges for conventional liquid membranes (10–100 μm) and the thinnest membranes in this work (190 nm) are also shown.

limited by $CO_2$ reaction rates even at high thicknesses of 10–100 μm due to the relatively slow reaction rates (Fig. 4b). The carbonic anhydrase enzyme enables more rapid carbonate reaction kinetics in liquid membranes, but the system will still become reaction-rate limited at low thicknesses. In contrast, transport in liquid water membranes relies on physical dissolution of $CO_2$ which has a rate constant on the order of $10^{10}\,s^{-1}$ so the system does not become dissolution rate limited until the thickness approaches the dimensions of water and $CO_2$ molecules[42]. Thus, water membranes are unique from other materials in that they can be fabricated as ultrathin layers, such as the 190-nm thick membranes demonstrated in this work, to increase permeance without encountering reaction rate limitations that limit permeance and reduce selectivity.

**Scalability and mixed gas testing of liquid water membranes**

Liquid water membranes can be implemented using any hydrophilic porous substrate with sufficiently small pores to maintain the water layer, making the approach inherently scalable. We demonstrated this principle using commercially available large-area membranes. Commercially produced hydrophilic polyvinylidene fluoride (PVDF) and polyethersulfone (PES) membranes were tested in gas separations after filling the pores with liquid water. PVDF and PES membranes were chosen because of the following: (1) they are commercially produced in large sheets (Fig. 5a) by a scalable phase inversion process, (2) they are modified to be hydrophilic by the commercial supplier, and (3) they have sub-micron pore sizes (Fig. 5b, c) which allow for stabilization of the liquid water layer.

Testing with commercially available hydrophilic membranes showed that liquid-water membranes demonstrate constant $CO_2$:$N_2$ selectivities around 40 regardless of the substrate used (Fig. 5d, left panel), in agreement with membrane transport theory and our results with ceramic membranes. For the unoptimized microns-thick commercial membranes used in this work, the $CO_2$ permeance was relatively low (5.1 GPU for PVDF and 6.1 GPU for PES). We attribute this low permeance to the large thickness of the water layer, which likely filled the entire thickness of the hydrophilic membrane and resulted in a long diffusion pathway for gases. The thickness of the water layer in the commercial membranes was approximately 100 μm (Supplementary Table 6), compared to 190 nm for the fabricated membranes.

To further assess performance in realistic conditions, multi-component gas testing under crossflow was conducted on a commercial PES membrane using a feed gas mixture of 90% $N_2$ and 10% $CO_2$ at a total pressure of 3.1 bar (Fig. 5d, right panel). The mixed-gas $CO_2$/$N_2$ selectivity was approximately 40, consistent with single-component measurements, indicating minimal interaction between the two gases in the aqueous phase.

Pressure stability measurements showed water displacement pressures of 1.5 bar for PVDF and 6.6 bar for PES. These pressures are sufficiently high for low-pressure gas-separation membrane applications but lower than those observed with alumina membranes, due to the larger, polydisperse pore sizes, which are vulnerable to gas breakthrough at lower pressures. Based on the wetting pressure, the largest pores in the PVDF and PES membranes are approximately 960 nm and 218 nm, respectively, far larger than the 40 nm pores used in fabricated membranes (Supplementary Table 6). Overall, results with commercial membranes show that large-scale porous substrates can be used as liquid water membranes, but further optimization to reduce the water layer thickness and membrane pore size is necessary to improve permeance and pressure tolerance.

## Discussion

This work introduces a new class of liquid water membranes that draws inspiration from the natural $CO_2$ uptake processes in plants. By fabricating membranes with hydrophilic sub-100-nm pores, we show that a stable water layer can be maintained at pressures exceeding 72 bar. Selectivity in the liquid water membrane is based primarily on solubility, where $CO_2$ is up to 40 times more permeable than other gases, such as $N_2$, due to its uniquely high solubility in water. Gas permeance can be increased by decreasing the water layer thickness without compromising selectivity, and membranes with 190-nm thick water layers show gas permeances exceeding 11,000 GPU while maintaining $CO_2$:$N_2$, $CO_2$:$CH_4$, and $CO_2$:$H_2$ selectivities of 40, 26, and 31, respectively. Membranes exhibit stable performance over eight days under dry, high-pressure conditions and maintain performance under high $CO_2$ partial pressures and mixed gas feed streams in crossflow. Moreover, commercial large-scale membranes can be used with liquid water, although further optimization is required to improve permeance and pressure tolerance.

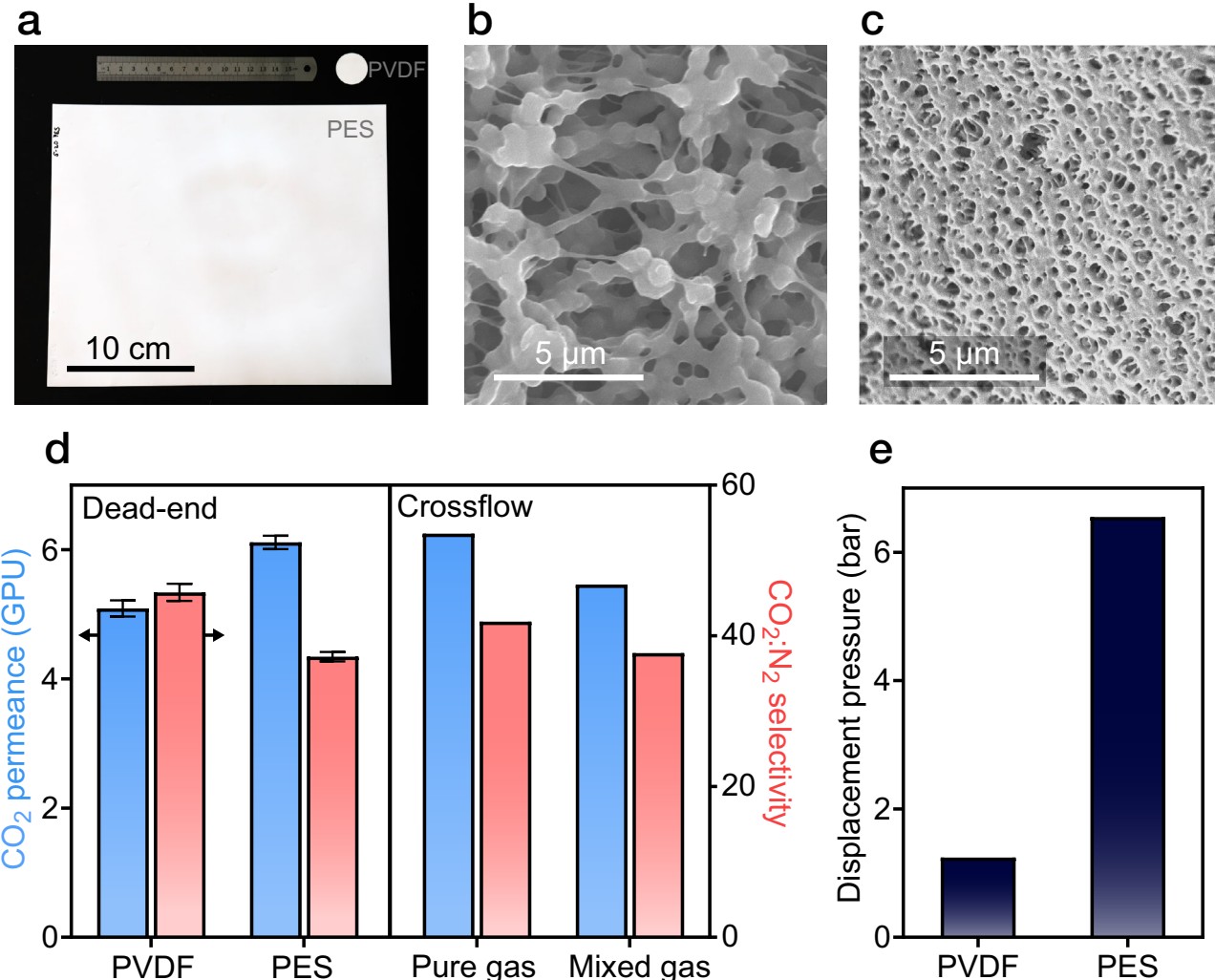

**Fig. 5 | Water membranes are scalable using commercial membrane substrates under single- and mixed-gas operation. a** Photograph of the hydrophilic polyvinylidene fluoride (PVDF) and polyethersulfone (PES) membrane samples. **b,c** Scanning electron micrograph of the top surface of the hydrophilic PVDF and PES membranes, respectively. **d** Measured $CO_2$ permeance and $CO_2$:$N_2$ selectivity of PVDF and PES membranes with trapped water layers. Permeance and selectivity are shown using single-component gas feeds in a dead-end cell (left panel) and both single-component and multicomponent gas feeds in a crossflow system (right panel; results shown only for PES). Error bars denote the mean ±1 s.d. for measurements from at least three distinct membrane samples. **e** Measured displacement pressures of water for PVDF and PES membranes.

Liquid water membranes show promise for addressing separation challenges in applications such as carbon capture, biogas processing, natural gas sweetening, and syngas upgrading. In these applications, the high $CO_2$ permeance, humidity tolerance, and pressure stability demonstrated in the liquid membranes introduced in this work may enable substantial improvements in cost, energy efficiency, and reliability. Unlike many conventional systems, water membranes are inherently compatible with humid or water-saturated gas streams, potentially eliminating the need for dehumidification processes common in many gas separation applications. Moreover, because water is non-toxic and environmentally benign, this approach offers a sustainable and potentially regenerable alternative to solvent-based or polymeric membranes.

Future efforts should prioritize scaling fabrication methods, evaluating long-term reliability, and assessing performance in full-scale process environments. We demonstrate operation at pressures up to 72 bar without breakthrough, but condensation or dissolution of contaminants in the water layer (e.g., $SO_2$, $H_2S$, and light hydrocarbons) could reduce surface tension and lower that threshold. Long-term stability of the water layer, particularly under dry or low-humidity conditions should also be examined. With continued development, liquid water membranes may enable the high-performance gas separations essential for a sustainable economy.

## Methods

### Liquid water membrane preparation

Membranes were prepared by pipetting 20 µL of deionized water onto a hydrophilic porous support and allowing the water to wick into the pores. Commercially available flat-sheet anodic alumina (InRedox, CO, USA), hydrophilic polyethersulfone (Sterlitech PES00347100), and polyvinylidene fluoride (Millipore HVLP02500) membranes were tested without modification. Membranes with a thin (less than 1 µm) hydrophilic layer were fabricated using anodic alumina substrates with varying pore diameters (20, 40, 80, and 120 nm) and a thickness of 50 µm (InRedox, CO, USA). The substrates were rinsed with acetone, ethanol, and isopropyl alcohol, then dried under nitrogen. The substrates were then hydroxylated using a UV ozone cleaner (Ossila, Sheffield, UK) for 5 mins and placed in a 10% $H_2O_2$ solution at 40 °C for 15 minutes before being exposed to 1H,1H,2H,2H-perfluorodecyltriethoxysilane at a temperature of 130 °C and a vacuum

pressure of 50 mTorr for 8 hours to create a hydrophobic layer of fluorocarbons on the anodic alumina membrane. All silane modifications were performed in the vapor phase using a chemical vapor deposition chamber (Supplementary Fig. 17). After hydrophobic modification, platinum was then deposited on the membranes using a magnetron sputter coater (Leica ACE600, Germany) with varying incident angles of 65° to 85° with a target distance of 50 mm, a current of 35 mA, and argon as the sputtering gas. The angle of sputtering determined the depth of the platinum layer using geometric calculations confirmed with experimental measurements and imaging. Platinum sputtered membranes were then exposed to 10 mM thioglycolic acid in isopropyl alcohol for 3 h to selectively modify the platinum layer to become hydrophilic. The membranes were then rinsed with isopropyl alcohol, dried with nitrogen gas, and placed on a hot plate set at 40 °C for a minimum of 3 h before performance testing.

### Displacement pressure measurements
The pressure at which the water layer is displaced from the pores of the membrane was determined by the measurement of abrupt changes in gas flow, up to $7 \times 10^5$ GPU for dry anodic alumina membranes (Supplementary Fig. 18), from the permeate side of the membrane under applied pressure. The high-pressure (up to 75 bar) membrane holder is shown in Supplementary Fig. 19. Membrane samples with a 13 mm diameter were loaded in the test cell, and pressure was applied using compressed nitrogen gas and held for at least 5 min at each pressure increment. Pressure was increased in increments of 0.7 bar until membrane failure was observed via rapid gas transport through the pores.

### Gas permeance testing
Gas transport across the membrane was measured using a membrane flow cell (In Redox, 304 stainless membrane holder) immersed in a DI water bath to maintain a temperature of $22 \pm 2$ °C (Supplementary Fig. 14). Single gas permeation testing was performed using $H_2$, $CO_2$, $CH_4$, $N_2$, and $O_2$ gases (Airgas Part HY UHP300, CD FG20, ME CP80, NI UHP300, and OX USP200 cylinders, respectively). Fabricated thin membranes were oriented in the membrane cell with the thin, hydrophilic layer facing the permeate side to prevent condensation and water accumulation on the feed side. Depending on the gas permeation rate, gas flux was measured using an electronic gas flow meter (Aalborg Digital Mass Flowmeter Part 503924), a bubble gas flow meter (Manual Bubble Flowmeter Part 20433-U), or by tracking the displacement of a water column on the permeate side. Transmembrane pressure was controlled by adjusting the feed-side pressure from 1 to 72 bar, while the permeate side was maintained at atmospheric pressure. Relative humidity was measured on the feed side using a high-pressure humidity sensor (Roscid Technologies HDR200). Humidity was controlled by bubbling the feed gas through 15 mL of water at 25–50 °C. Gas flux was quantified in units of $cm^3$ (STP)/(cm²·s), and permeance is shown in gas permeation units (GPU), where 1 GPU = $10^{-6}$ $cm^3$ (STP)/(cm²·s·cmHg). Multicomponent gas testing under crossflow conditions was conducted using $CO_2$ and $N_2$ mass flow controllers on the feed side, with helium used as a sweep gas on the permeate side (Maxwell Robotics, Mixed Gas Permeation Analyzer, Austin, Texas, USA). Humidity was introduced to both the feed and permeate streams using water vaporizers and was maintained above 90% relative humidity, as measured by hygrometers installed on both sides of the membrane. Gas permeability and selectivity were determined by gas chromatography analysis of the permeate stream.

### Membrane characterization
Membrane hydrophobicity was characterized with an optical tensiometer (Biolin Scientific, AZ, USA) using the sessile drop method with a 10 μL water droplet. The top surface of the membrane was imaged using field-emission scanning electron microscopy (SEM) from a JEOL

JSM-7401F (Tokyo, Japan). The membrane cross-section and thickness of the platinum layer were imaged using scanning transmission electron microscopy (STEM) (FEI Co. Talos F200X 200keV). X-ray photoelectron spectroscopy (XPS) (Kratos Supra, Kratos Analytical, UK) was performed by irradiating samples with a monochromatic Al K beam with a source energy of 1486.69 eV, and an X-ray beam power and resolution of 225 W.

## Data availability
Source data are provided with this paper. Additional data supporting the findings of this study are available in the Supplementary Information and from the corresponding author upon request. Source data are provided with this paper.

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

## Acknowledgements

K.P.L. appreciates support from the National Aeronautics and Space Administration (NASA) via the NASA Space Technology Graduate Research Opportunity (NSTGRO) fellowship. A.P.S. gratefully acknowledges support from the U.S. National Science Foundation under CAREER Award No. CBET-2442780. We thank Dragan Mejic at the University of Colorado Boulder Chemical Engineering Instrument Shop for his assistance in manufacturing components of the high-pressure gas permeation cell. We also thank Praveen Kumar at Colorado School of Mines for assistance with transmission electron microscopy. This work

was authored in part by the National Laboratory of the Rockies (NLR), operated by Alliance for Sustainable Energy, LLC, for the U.S. Department of Energy (DOE) under Contract No. DE-AC36-08GO28308. This work was supported by the Transformational Laboratory Directed Research and Development (TLDRD) Program at NLR. The views expressed in the article do not necessarily represent the views of the DOE or the U.S. Government. The U.S. Government retains, and the publisher, by accepting the article for publication, acknowledges that the U.S. Government retains a nonexclusive, paid-up, irrevocable, worldwide license to publish or reproduce the published form of this work, or allow others to do so, for U.S. Government purposes.

## Author contributions

K.P.L. designed and conducted membrane fabrication, characterization, and permeation experiments and analyzed the results. A.P.S. conceived the project and provided scientific supervision throughout the study. M.A., Y.T., and P.O.S. performed permeation experiments and contributed to the analysis and interpretation of the transport data with supervision from M.P., A.R., and A.P.S. M.S.M. fabricated membranes, conducted fabrication-related characterization and analyses, and contributed to manuscript writing. J.N.S. performed XPS measurements and analysis. S.R.N. conducted SEM imaging and analysis. K.P.L., M.S.M., M.A., Y.T., J.N.S., S.R.N., P.O.S., M.P., A.R., and A.P.S. discussed the results. K.P.L., M.S.M., and A.P.S. wrote the manuscript; all authors revised and approved the final version.

## Funding

## Competing interests

M.S.M., M.A., J.N.S., S.R.N., Y.T., P.O.S., M.P., and A.R. declare no competing financial interests. K.P.L and A.P.S are cofounders, board members, and officers of Osmopure Technologies Inc., and they hold equity in the materials covered in this work.
