## [Transparent Peer Review file · Nature Communications]

Water as a gas separation membrane

Corresponding Author: Professor Anthony Straub

Version 0:

Reviewer comments:

Reviewer #1

(Remarks to the Author)

This is my second review of this manuscript in a Nature family journal. My comments on my first review for nature sustainability were as follows, where I provide in italics additional comments based on this mildly revised manuscript: This paper reports the formation of a supported/nanoconfined water membrane and its performance in various CO₂ separations. The approach employed a commercial planar anodic alumina nanoporous support that was hydrophobized with an organosilane followed by surface derivatization with platinum and a hydrophilic thiol to render the immediate surface hydrophilic/superhydrophilic? to a depth of about 190-nm. The rather uniform hydrophilic pores spontaneously fill with water due to capillary condensation and are stabilized to high pressure according to the Young Laplace equation. The selectivities of the corresponding membranes based on single gas permeance measurements were reported to be CO₂:N₂, CO₂:CH₄, and CO₂:H₂ equal to 40, 26, and 31, respectively, with permeances as high as 11,600 GPU (presumably at high pressure operation). The membranes are claimed to have performances exceeding the Robeson upper and to outperform state-of-the-art materials reported in the literature. Conceptually the idea of forming a very thin stabilized water-based membrane was already reported in Fu, Y. et al. Ultra-thin enzymatic liquid membrane for CO₂ separation and capture. Nat Commun 9, 990 (2018) (ref. 19 in the current paper). In that work a planar anodic alumina nanoporous support was first modified with a mesoporous silica film to unify the pore size at about 8-nm. The pore surfaces were hydrophobized with an organosilane, as in the present work, followed by a remote oxygen plasma treatment to hydroxylate the pores and render them hydrophilic to a depth of about 20-nm. As in the current paper water would condense in the pores and be stabilized toward blowout and evaporation. Unlike the present work, Fu et al used the stabilized ultra-thin (~20-nm) water membrane to accommodate/stabilize carbonic anhydrase, an enzyme that selectively dissolves CO₂. The enzymatic liquid membrane separated CO₂ at room temperature and atmospheric pressure at a rate of 2600 GPU with CO₂/N₂ and CO₂/H₂ selectivities as high as 788 and 1500, respectively. In the present work the authors do not compare their membranes to the Fu et al membranes and incorrectly state that these (Fu et al) systems rely on reaction mechanisms that become rate-limiting at low membrane thicknesses, thereby constraining CO₂ permeance (It is unclear to me what is even meant by that statement). To elaborate on this point further, the authors ignore the various mechanisms by which CO₂ dissolves in water. If the authors claim their work to be bio-inspired they should recognize that in nature CO₂ dissolves in water principally by the reversible formation of carbonic acid. Carbonic anhydrase significantly speeds up this chemical where carbonic acid then dissociates to form bicarbonate ions which diffuse through water much more rapidly than gaseous CO₂ (implied in the present manuscript). Carbonic anhydrase (CA) catalyzed dissolution of CO₂ is one of the fastest enzymatic reactions known. A single molecule of CA can catalyze hundreds of thousands to over a million reactions per second, so contrary to the statement made in this manuscript,

“However, the gas permeance of supported liquid membranes is often limited by low gas diffusivity, difficulty in making thin liquid layers^{15,17}, and slow reaction kinetics in certain liquids that rely on chemical reactions with CO₂^{18,19}.”

CA-based CO₂ dissolution will not become reaction limited until the water thickness is on the scale of the enzyme itself. Beyond the questionable conceptual novelty of their stabilized water membrane and the inadequate comparison of results to ‘state-of-the-art’ membranes, i.e. the comparison with Fu, Y. et al. membranes, the idea of just using water as the selective layer is appealing for its simplicity and low cost. Unfortunately, the need for/use of Pt (to achieve very thin water membranes) detracts from the appeal and the commercial potential.

No matter the ultimate fate of this paper, mixed gas separations should be reported. Also a schematic of the measurement rig should be provided along with the operating conditions, for example, what are the transmembrane pressures?

Reviewer #2

(Remarks to the Author)

This is a well-written paper that requires only minor revision before it can be accepted for publication. I have the following comments:

1. This paper that describes a clever way to make “water” membranes for gas separations. The method used to make very thin water membranes is especially interesting. The authors may want to incorporate the Supplementary Figure 1 into the main article, I believe readers will be drawn to it.
2. Stability of the water phase is a major concern, as the authors discuss. Pressure stability is one challenge, for which the most straightforward solution is to keep operating pressures well below the breakthrough pressure. How likely is that the feed contains contaminants that can change the surface tension of the water?
3. The second stability issue is loss of the water phase, this may be the hard one.
4. Figure 1 i shows stability under pressurized conditions (why not plot actual permeance rather than relative permeance?), but the permeate side contains a droplet of water to avoid loss of water. Have experiments been carried out without that droplet? A requirement to keep the permeate side humidified is an obstacle for large scale implantation. The same has been proven true for facilitated transport membranes.
5. In the stability test the feed side is close to being dry. What would happen if the feed contains meaningful amounts of water vapor? The water phase will “grow” and perhaps the full feed surface of the membrane (which is hydrophilic) will be coated with a film of water? This would reduce permeances.
6. In the Discussion section the stability issue is missing from the future efforts language. It deserves to be included here.

Reviewer #3

(Remarks to the Author)

This manuscript presents a conceptually novel approach to gas separation using liquid water confined in sub-hundred-nanometer pores as the selective medium. The authors demonstrate that such membranes has high CO₂ permeance while maintaining high selectivity and can operate under high pressure and low humidity conditions. They further show that the separation mechanism follows a solubility-driven solution–diffusion model, where CO₂ selectivity arises from its much higher solubility in water relative to other gases. I think the concept is overall technically sound, elegantly simple, and quite novel. The mechanistic interpretation is coherent and supported by both theory and data. However, several aspects of the analysis and presentation require clarification and additional discussion, particularly regarding assumptions about pore geometry, active area normalization, and the completeness of the scale-up demonstration. That said, I'm in favor of the publication of this work after major revision.

Major comments

1. Reported permeance values are normalized assuming 12% membrane porosity. Given the tapered AAO pores, I am concerned that the true water filled area may deviate from this number significantly.
2. The pore size distribution in an AAO membrane is usually not uniform. The largest pores will actually determine the onset of water being pushed out. I think the authors need to provide a quantitative pore-size histogram. Their pressure stability analysis may also need to be adapted accordingly.
3. The scale-up experiments showcased the transferability of the concept but lack sufficient characterization. The authors should discuss the reasons for the lower permeance and pressure stability compared to the AAO tests. Even a qualitative discussion would strengthen this section substantially and may also provide guidelines for future works in this direction.
4. The model assumes negligible gas-phase resistance. I wonder how true this is when the membrane is only partially filled with water and the gas needs to go through long and small pores.
5. Industrial gas streams may contain contaminants (e.g., SO_x, hydrocarbons, etc), which can change water properties (even with a small amount). A discussion on the potential impacts and mitigation would enhance the paper's practical relevance.
6. The manuscript demonstrates 8-day stability under <1 % RH feed, but it would be more informative to estimate the water evaporation rate and expected drying timescale based on Eq. (5) and diffusion of water vapor through the gas phase. Even a simple order-of-magnitude estimate would help quantify long-term operational stability.

Minor comments

7. In Eq. (4), theta is defined as the intrinsic contact angle. In fact, the relevant parameter for gas displacing water should be receding contact angle. The authors should note that the static angle that they measured can only serve as an approximation of the receding angle.
8. In Eq. (6), r_c was used to represent minimum radius of curvature while already appearing in Eq. (5) to represent general radius of curvature. The authors should use a different symbol in this case.
9. GPU should be written in full in the abstract for broader accessibility.
10. Please provide the error range for reported membrane thickness

Reviewer #4

(Remarks to the Author)

Reviewer #5

(Remarks to the Author)

This study is well done and presents an innovative new concept that is worthy of consideration and further development. It presents a simple, nature-inspired idea: use a thin layer of water held inside tiny, hydrophilic pores as a membrane for separating gases based on their differential solubility in water. The authors show very fast CO₂ transport with solid selectivity over common gases (N₂, CH₄, H₂), good stability under high pressure, a week-long continuous run, and an initial path to scale using off-the-shelf porous supports. The idea of using water in a supported liquid membrane format is interesting and can potentially be a disruptive approach to CO₂ separations. However, there are a few items that we would like the authors to address so that the data presented is in context and easy to follow and so that the readers can see the pros and cons of the approach and this study.

1. The biggest challenge I have is with the equivalency that the authors are making with plant systems in figure 1 and in the intro parts. This is a little bit of a stretch, while the stomata are indeed water filled they are hardly nanosized (more like micron- sized) and there is no concept of nanoconfinement there. If the authors want to then delve into nanoscopic confinement of water and gases dissolving in it then we approach the controversial topic of whether membrane proteins transport gas. See for example <https://doi.org/10.1002/cphc.201100034>. I think this controversy is still not resolved in the field. Further even if we talk about membrane proteins and transport through them, the mechanism is not strictly solution diffusion perhaps because there is single file diffusion in many such channels and this makes it hard to equate bulk phenomena like solution diffusion to it as discussed. I would urge the authors to soften this comparison as it is not really needed for the nice set of results they already have.
2. Along the same lines, the title seems too general "Water as a gas separation membrane" while catchy is not strictly true. Water has to be confined or held within other membrane materials for the work to be possible so the authors could think about a more accurate title? I think the authors don't emphasize enough that this concept is most analogous to supported liquid membrane, but they have turned the concept on its head and use water instead of solvents.
3. Lines 73-75 "Plants, in particular, have evolved to uptake CO₂ by dissolving it in water-filled nanochannels that line the walls of leaf mesophyll cells (Figure 1a)23–25." This according to my understanding is not correct but I am happy to look at any evidence the authors present.
4. Line 211-212 also other places. "Gas flux was normalized to the membrane porosity of 12% so that reported values are representative of the gas transport rate through the water-filled active pore area." I am not sure this is a fair comparison. The area should be the membrane area. Do other supported liquid membranes do this?
5. Section "Permeance and selectivity performance in gas separations". The discussion of selectivity and permeability in this paper and its comparison with conventional membranes is quite confusing and confused here. It seems like that the overall solution diffusion model is still valid here, the only advantage is that the thickness of the actual active part of the membrane can be minimized and is possible with this particular approach as opposed to traditional membranes or other supported liquid membranes. We think this section should describe this in a straightforward without convoluting arguments in favor of the current membrane.
6. We appreciate that the authors tried out conventional membrane platforms in addition to using anodized aluminum oxide membranes and have pointed out the challenges in scaling this technology based on the vast difference in performance seen.
7. Line 310: There is an error in numbering Figure 4b and 4c
8. All data in the paper are on single gas experiments. While we understand the challenges of conducting mixed gas experiments on the current (really new) system, there should be some discussion on the current system's expected performance with realistic mixed gas streams.

Version 1:

Reviewer comments:

Reviewer #1

(Remarks to the Author)

- 1) My overall impression is that the analogy to water uptake and transport in trees is over hyped. Water is transported in trees through combined transpiration (evaporation) and capillary stabilization in small pores like aquaporin channels. Here evaporation is to be avoided to maintain the water in the membrane pore channels.
- 2) The novelty is limited in that there are review articles concerning supported liquid membranes for gas separation, e.g., ref.35, where it is stated, "capillary forces and are primarily responsible for solvent persistence; therefore, they contribute significantly to SLM stability by preventing the solvent from being escaped especially under high differential gas pressures.129–134 The capillary forces and hence the SLMs stability can be significantly increased by immobilizing the solvent in micropores or nanopores rather than mesopores or macropores, which prevents the solvent from escaping.
- 3) In Fig. 3, the authors selected only a limited set of membranes to which to compare their membranes and claim that they exceed the existing Robeson upper bound. Other supported liquid membranes like those described in ref. 28 along with

carbon molecular sieves, e.g. Rahimalimamaghani et al. (Industrial & Engineering Chemistry Research 2023, 62, 45, 19116-19132) have CO₂/N₂ selectivities in excess of 150 and should be plotted in Fig. 3.

- 4) I maintain that to be published in a top tier journal like Nature Comm, a membrane article should report mixed gas permeabilities. That could be done using simply GC analysis of the permeate and retentate streams compared to the feed stream.

- .

Reviewer #2

(Remarks to the Author)

This reviewer (#2) was not part of (and did not see) the first round of reviews. The second round reviews by the other reviewers is quite extensive and covers topics not addressed by me. I have decided to stay out of those discussions.

The author's response to my comment 6 raises a new question: All permeation experiments were carried out with the hydrophilic side facing the permeate? I must have missed it if there is any mention of this in the manuscript, but it certainly has to be disclosed. Furthermore, this membrane orientation works for single component permeation experiments but not for mixture experiments: concentration polarization in the pore section above the water phase will reduce selectivity and permeance. Something to be aware of when mixture experiments are carried out (which as other reviewers have mentioned is a required next step).

It will take a minor revision to address my comment above, after which the manuscript can be published as far as this reviewer is concerned.

Reviewer #3

(Remarks to the Author)

The authors have addressed all my comments in the previous round. I now support the publication of this work in Nat. Comm.

Reviewer #4

(Remarks to the Author)

Reviewer #5

(Remarks to the Author)

I am satisfied with the revisions that the authors have made in response to reviewer comments.

Version 2:

Reviewer comments:

Reviewer #1

(Remarks to the Author)

The authors have addressed my concerns.

Reviewer #2

(Remarks to the Author)

I thank the authors for the responses to the reviewers. The manuscript is acceptable for publication.

Response to Reviewers

We would like to thank the five reviewers for their thoughtful suggestions and constructive feedback that helped us to improve the manuscript. Below is a point-by-point response to comments from the reviewers and a detailed description of the revisions made to the manuscript. In our response, black italic type represents the exact comments from the reviewer, blue text represents new text added to the revised manuscript, and red strikethrough text represents text deleted during the revision. Lines listed refer to the revised manuscript.

REVIEWER COMMENTS

Reviewer #1

Comment 1: *This is my second review of this manuscript in a Nature family journal. My comments on my first review for nature sustainability were as follows, where I provide in italics additional comments based on this mildly revised manuscript:*

Our Response: We thank the reviewer for taking the time to review our manuscript for a second time. We appreciated your thorough earlier review for *Nature Sustainability* and edited our current submission after consideration of those comments. We have also addressed the additional comments in this response and made substantial changes to the manuscript. We are grateful for your continued feedback, which has further strengthened this work.

Comment 2: *This paper reports the formation of a supported/nanoconfined water membrane and its performance in various CO₂ separations. The approach employed a commercial planar anodic alumina nanoporous support that was hydrophobized with an organosilane followed by surface derivatization with platinum and a hydrophilic thiol to render the immediate surface hydrophilic/superhydrophilic? to a depth of about 190-nm. The rather uniform hydrophilic pores spontaneously fill with water due to capillary condensation and are stabilized to high pressure according to the Young Laplace equation. The selectivities of the corresponding membranes based on single gas permeance measurements were reported to be CO₂:N₂, CO₂:CH₄, and CO₂:H₂ equal to 40, 26, and 31, respectively, with permeances as high as 11,600 GPU (presumably at high pressure operation). The membranes are claimed to have performances exceeding the Robeson upper and to outperform state-of-the-art materials reported in the literature. Conceptually the idea of forming a very thin stabilized water-based membrane was already reported in Fu, Y. et al. Ultra-thin enzymatic liquid membrane for CO₂ separation and capture. Nat Commun 9, 990 (2018) (ref. 19 in the current paper). In that work a planar anodic alumina nanoporous support was first modified with a mesoporous silica film to unify the pore size at about 8-nm. The pore surfaces were hydrophobized with an organosilane, as in the present work, followed by a remote oxygen plasma treatment to hydroxylate the pores and*

render them hydrophilic to a depth of about 20-nm. As in the current paper water would condense in the pores and be stabilized toward blowout and evaporation. Unlike the present work, Fu et al used the stabilized ultra-thin (~20-nm) water membrane to accommodate/stabilize carbonic anhydrase, an enzyme that selectively dissolves CO₂. The enzymatic liquid membrane separated CO₂ at room temperature and atmospheric pressure at a rate of 2600 GPU with CO₂/N₂ and CO₂/H₂ selectivities as high as 788 and 1500, respectively. In the present work the authors do not compare their membranes to the Fu et al membranes and incorrectly state that these (Fu et al) systems rely on reaction mechanisms that become rate-limiting at low membrane thicknesses, thereby constraining CO₂ permeance (It is unclear to me what is even meant by that statement).

Our Response:

We thank the reviewer for this comment. We note that this helpful comment was made on the previous submission of our manuscript to a different journal, and we revised our submission to *Nature Communications* to address some of these points. We have also made further revisions to clarify the differences in membrane and transport properties between our work, which focuses on membranes using only water as the selective layer, and the prior publication by Fu et al, which showed impressive performance by leveraging enzymes.

The scope of the current study focuses on membranes where pure water serves as the selective layer for gas separation, without the use of enzymes or reactive additives. Our goal is to understand and characterize the intrinsic transport behavior of gases through stabilized water layers. The main contributions of this work are as follows:

- We demonstrate that gas transport through thin water layers follows a simple solution-diffusion model, allowing direct comparison between theory and experiment. Following the model, selectivity primarily depends on gas physical solubility in water.
- We show that membrane performance is maintained under elevated feed pressures (up to 72 bar), leveraging capillary forces to stabilize the liquid water phase against displacement.
- We provide a framework to tune and stabilize the water layer thickness, down to submicron scales, to reach CO₂ permeances up to 11,600 GPU while maintaining a CO₂:N₂ selectivity of 40, which is desirable for some gas separation applications.
- We demonstrate long-term stability over 8 days under dry conditions, enabled by a reduction in the equilibrium vapor pressure of water via nanoscale curvature (Kelvin effect).

We view our work as complementary to the pioneering study by Fu et al. (*Nat. Commun.* 2018), which introduced an ultrathin enzymatic water membrane supported in anodized alumina. That study reported extremely high CO₂ selectivity and permeance using carbonic anhydrase, and differed from our current work in several ways:

- Our membranes rely primarily on physical dissolution of CO₂ in water. Transport in the water membranes follows the solution-diffusion model using parameters for physical

dissolution and diffusion of CO₂ in water, allowing for direct prediction of permeance based on measured water layer thickness. For example, our 190 nm water layer yields a theoretical CO₂ permeance of 10,400 GPU, which closely matches our measured value for 11,600 GPU.

- The membranes of Fu et al. rely on catalytic reactions with carbonic anhydrase. These membranes used a thinner water layer approximately 20 nm thick, but the observed permeance (~2,600 GPU) was lower than the 100,000 GPU predicted by solution-diffusion. They attributed this to rate limitations associated with enzymatic catalysis during CO₂ hydration and regeneration steps, making CA reaction kinetics the controlling factor in their system: “given the known CO₂ permeability in pure water¹⁵ ($210 \times 10^{-9} \text{ cm}^3 \text{ (STP) cm sec}^{-1} \text{ cm}^{-2} \text{ cm}^{-1} \text{ Hg}^{-1}$), the permeance of the designed 20-nm-thick water-membrane should be of $210 \times 10^{-9} \text{ cm}^3 \text{ (STP) cm sec}^{-1} \text{ cm}^{-2} \text{ cm}^{-1} \text{ Hg}^{-1}$ divided by the thickness, which is $0.1 \text{ cm}^3 \text{ s}^{-1} \text{ cm}^{-2} \text{ cm}^{-1} \text{ Hg}^{-1}$, or 10^5 GPU. Now, since a CO₂ permeance of 10^5 GPU is much larger than the permeance observed (Fig. 7f), it follows that the CA-catalyzed steps I or III are rate limiting.”
- As a result of the different transport mechanisms, membranes in the two papers show different gas separation performance: the membranes of Fu et al. are ultrasensitive, whereas the membranes shown in this work have very high permeances. The membranes in Fu et al. achieved CO₂ selectivities much higher than the work here (788 vs. 40) since the carbonic anhydrase facilitated more rapid transport of CO₂ as compared to other gases. However, since the membranes in our current work rely only on physical dissolution in water, very high permeances can be achieved with thin water layers as compared to the membranes of Fu et al. (11,600 GPU vs. 2,600 GPU). We note that each membrane type might find use for distinct applications. For example, lower selectivity can be acceptable or even preferred for point source carbon capture. Thus, the pure water membrane and enzymatic membranes may each be appropriate for different applications requiring either high throughput or selectivity.
- The membranes shown in the work comprise nanopores filled with water and use capillary forces to withstand very high pressures (up to 72 bar) that follow predictions from Young-Laplace theory (Figure 2e). In contrast, the membranes of Fu et al. were not tested at high differential pressures (up to 0.48 bar).
- We show that water membranes can use commercially available hydrophilic substrates and still operate at reasonable pressure (Figure 5), which helps to demonstrate scalability of the process. The water membranes do not intrinsically require enzymes or other chemical additives.

Other recent studies, such as Sandru et al. (*Science*, 2023), have also leveraged water in membrane-based CO₂ separations. However, those systems rely on facilitated or reactive transport within hydrated polymers. While water plays an essential role in their performance, it acts as part of a more complex, chemically reactive transport mechanism. In contrast, our work isolates the transport of water alone, allowing for a comprehensive mechanistic interpretation and showing water’s potential as a stable separation layer.

We have revised the text as follows:

Line 80 – 85: “Despite the promise of water as a medium for gas separations, relatively few studies have explored water in an engineered gas separation membrane.⁵ Prior work has incorporated carbonic anhydrase, ionic liquids, or other reactive chemicals into an aqueous layer to enable facilitated transport^{3,28,29}, ~~and none have demonstrated it’s unique advantages in CO₂ transport and pressure tolerance~~ but membranes that mimic the high-pressure tolerance of capillary-stabilized water observed in leaves to improve CO₂ separations remain unexplored.”

In response to the comment on the high pressure in which our permeances were obtained, it should be clarified that permeance (GPU) is a measure of flux normalized to pressure and membrane area:

$$1 \text{ GPU} = 3.35 \times 10^{-10} \frac{\text{mol}}{\text{m}^2 \cdot \text{s} \cdot \text{Pa}}$$

We find that gas permeance in the water membranes does not depend on pressure because flux increases linearly with pressure (Figure 2f). The permeance measured in this work aligns with the solution-diffusion model for gas transport through a stabilized liquid film and depends only on the solubility and diffusivity of the gas in water, the thickness of the water layer, and the porosity of the support (Figure 2f-h). We note that the ability to operate with high CO₂ partial pressure without a loss in permeance or selectivity is an advantage of these membranes as compared to other current technologies, such as ionic liquids or polymers.

Comment 4: *To elaborate on this point further, the authors ignore the various mechanisms by which CO₂ dissolves in water. If the authors claim their work to be bio-inspired they should recognize that in nature CO₂ dissolves in water principally by the reversible formation of carbonic acid.*

Our Response:

We appreciate the reviewer’s comment and agree that in biological systems, CO₂ physically dissolves in water and then is converted to bicarbonate and protons using carbonic anhydrase. Our membranes do not mimic this pathway, since the short diffusion path and lack of enzymes means CO₂ conversion to bicarbonate negligibly contributes to the overall transport. Physical dissolution being the dominant pathway in our system is consistent with our measured selectivities, which match those expected from the physical solubility ratios of CO₂ relative to N₂, O₂, H₂, and CH₄ in water. If we had substantial conversion of CO₂ to bicarbonate in our system, we would expect higher selectivities than those observed.

In this work, the biological inspiration lies in the membrane’s structural design that allows for water to be stabilize in nanoscale capillaries at high pressures comparable to those of tall trees like the Coastal Redwood. Our membranes use hydrophilic nanopores to maintain a

stable water layer through capillary forces under those elevated pressures. This design is analogous to the water-filled channels in plant leaves within the walls of mesophyll cells, which have pores on the order of 10 nm and tolerate pressure differences higher than 50 bar (Wang et al., 2020, *Science Advances*, Ref. 28). We have revised the main text of the manuscript to clarify, as shown in our previous response:

Line 80 – 85: “Despite the promise of water as a medium for gas separations, relatively few studies have explored water in an engineered gas separation membrane.⁵ Prior work has incorporated carbonic anhydrase, ionic liquids, or other reactive chemicals into an aqueous layer to enable facilitated transport^{19,20,21}, ~~and none have demonstrated its unique advantages in CO₂ transport and pressure tolerance~~ but membranes that mimic the high-pressure tolerance of capillary-stabilized water observed in leaves to improve CO₂ separations remain unexplored.”

Comment 4: *Carbonic anhydrase significantly speeds up this chemical where carbonic acid then dissociates to form bicarbonate ions which diffuse through water much more rapidly than gaseous CO₂ (implied in the present manuscript). Carbonic anhydrase (CA) catalyzed dissolution of CO₂ is one of the fastest enzymatic reactions known. A single molecule of CA can catalyze hundreds of thousands to over a million reactions per second, so contrary to the statement made in this manuscript, “However, the gas permeance of supported liquid membranes is often limited by low gas diffusivity, difficulty in making thin liquid layers^{15,17}, and slow reaction kinetics in certain liquids that rely on chemical reactions with CO₂^{18,19}.” CA-based CO₂ dissolution will not become reaction limited until the water thickness is on the scale of the enzyme itself.*

Our Response: We thank the reviewer for bringing up the topic of reaction rate limitations. We fully agree that carbonic anhydrase is among the most efficient enzymes known, with the ability to catalyze one million reactions per second. However, we would like to clarify the context of our original statement and address the conditions which even such rapid kinetics may still become rate-limiting. In the revised main text, we have included additional analysis using the Damköhler number (Equation 7 in the main text and included below) to estimate the transition between reaction limited and diffusion-limited regimes in thin liquid membranes. Specifically, we calculate that even with a high catalytic rate constant ($k_r \sim 200,000 \text{ s}^{-1}$), reaction kinetics can begin to limit transport for water layers thinner than approximately 100 nm (Figure 4b, copied below), assuming typical diffusivities for bicarbonate in water. We also note that Fu et al. reported a measured CO₂ permeance for a 20 nm carbonic anhydrase containing membrane that was significantly lower than the value expected from physical diffusion alone, which they themselves attributed to rate-limiting enzymatic steps: “given the known CO₂ permeability in pure water¹⁵ ($210 \times 10^{-9} \text{ cm}^3 \text{ (STP) cm sec}^{-1} \text{ cm}^{-2} \text{ cm}^{-1} \text{ Hg}^{-1}$), the permeance of the designed 20-nm-thick water-membrane should be of $210 \times 10^{-9} \text{ cm}^3 \text{ (STP) cm sec}^{-1} \text{ cm}^{-2} \text{ cm}^{-1}$

$^1 \text{Hg}^{-1}$ divided by the thickness, which is $0.1 \text{ cm}^3 \text{ s}^{-1} \text{ cm}^{-2} \text{ cm}^{-1} \text{ Hg}^{-1}$, or 10^5 GPU . Now, since a CO_2 permeance of 10^5 GPU is much larger than the permeance observed (Fig. 7f), it follows that the CA-catalyzed steps I or III are rate limiting.” This supports the idea that even reaction rates as fast as carbonic anhydrase can become the rate limiting step in thin water layers.

$$Da = \frac{k_r l^2}{D} \quad (7)$$

Fig. 4 | Sorption capacity and dissolution kinetics of CO₂ in water. a, CO₂ permeability (barrer) as a function of CO₂ partial pressure in supported ionic liquid and water membranes. At high CO₂ partial pressure, ionic liquids with high CO₂ solubility can become saturated resulting in lower CO₂ permeability in supported ionic liquid membranes. Water maintains constant CO₂ permeability at high pressure because CO₂ transport is driven by physical dissolution^{17,20,53}. **b**, Critical thickness at which a system transitions from diffusion- to reaction-limited CO₂ transport (Da = 1) for different sorption and reaction processes: CO₂ reaction to form carbonate, CO₂ reaction with liquid amine, and limiting reaction rate of carbonic anhydrase enzyme^{20,54–56}. Gas dissolution kinetics in water are rapid, and the equivalent critical thickness of water is below 1 nm. Typical thickness ranges for conventional liquid membranes (10–100 μm) and the thinnest membranes in this work (190 nm) are also shown.

Comment 5: *Beyond the questionable conceptual novelty of their stabilized water membrane and the inadequate comparison of results to ‘state-of-the-art’ membranes, i.e. the comparison with Fu, Y. et al. membranes, the idea of just using water as the selective layer is appealing for its simplicity and low cost. Unfortunately, the need for/use of Pt (to achieve very thin water membranes) detracts from the appeal and the commercial potential.*

Our Response: We thank the reviewer for this comment and acknowledge that it was also raised on a previous submission to another journal. In response, we have significantly

revised the manuscript to better clarify both the novelty and the broader relevance of our approach.

To address the concern regarding the use of platinum in our proof-of-concept membranes, we added data from commercially available, lower-cost porous substrates such as PVDF and PES (Figure 5, copied below). These experiments show that stable water membranes with high CO₂ selectivity can also be achieved using inexpensive materials, suggesting a path toward scalable and commercially viable designs, though additional optimization is still needed to increase permeance and pressure tolerance. We note that platinum is not fundamentally required for the thin membrane design, but it was a convenient and easy-to-sputter material for our proof-of-concept.

Fig. 5 | Water membranes are scalable using commercial membrane substrates. **a**, Photograph of the hydrophilic polyvinylidene fluoride (PVDF) and polyethersulfone (PES) membrane samples. **b,c**, Scanning electron micrograph of the top surface of the hydrophilic PVDF and PES membranes, respectively. **d**, Measured CO₂ permeance and CO₂:N₂ selectivity of PVDF and PES membranes with trapped water layers. Error bars denote the mean \pm 1 s.d. for measurements from at least three distinct membrane samples. **e**, Measured displacement pressures of water for PVDF and PES membranes.

Comment 6: *No matter the ultimate fate of this paper, mixed gas separations should be reported. Also a schematic of the measurement rig should be provided along with the operating conditions, for example, what are the transmembrane pressures?*

Our Response: We appreciate this suggestion and have also included a schematic of the measurement rig in the Supporting Information (included below).

Supplementary Fig. 14 | Experimental setup for gas permeance and long-term stability testing.

We agree that mixed gas testing is important to demonstrate realistic performance of this process. However, mixed gas testing for our membranes would require a system that can sweep gases on both sides of the membrane, allow for humidification, and measure separation performance. Very few instruments like this exist globally, and we are unfortunately currently unable to conduct mixed gas testing appropriate for this system. We hope to have follow on studies with mixed gases.

It should be noted that we did test gas permeability as a function of pressure (Fig. 4a) and observed no change at high partial pressures. Similarly, we anticipate that, since aqueous gas solubility is not expected to change with additional dissolved gases, mixed gas separation performance will be similar to the results obtained with pure gases. The water layer is also not expected to show effects from plasticization or competitive sorption, both of which are phenomena that cause mixed gas selectivities to fall short of those with pure gas streams.

We have revised the text to note the importance of mixed gas testing in future studies.

Line 430 – 440: “Moreover, because water is non-toxic and environmentally benign, this approach offers a sustainable and potentially regenerable alternative to solvent-based or polymeric membranes. Future efforts should prioritize scaling fabrication methods, evaluating long-term reliability, and assessing performance in full-scale process environments. We demonstrate operation at pressures up to 72 bar without breakthrough, but condensation or dissolution of contaminants in the water layer (e.g., SO₂, H₂S, and light hydrocarbons) could reduce surface tension and lower that threshold. Although water is not prone to competitive sorption effects, future work should evaluate membrane performance under realistic multicomponent gas feeds and further assess long-term stability of the water layer, particularly under dry or low-humidity conditions. With continued development, liquid water membranes may enable the high-performance gas separations essential for a sustainable economy.”

We thank the reviewer for their question on transmembrane pressures and agree this requires additional explanation. We have revised the text as follows:

Line 475 – 479: “Depending on the gas permeation rate, gas flux was measured using an electronic gas flow meter (Aalborg Digital Mass Flowmeter Part 503924), a bubble gas flow meter (Manual Bubble Flowmeter Part 20433-U), or by tracking the displacement of a water column on the permeate side. Transmembrane pressure was controlled by adjusting the feed-side pressure from 1 to 72 bar, while the permeate side was maintained at atmospheric pressure.”

Reviewer #2:

Comment 1: *This is a well-written paper that requires only minor revision before it can be accepted for publication. I have the following comments:*

Our Response: We thank the reviewer for the positive assessment and recommendation. We have revised the manuscript to address their thoughtful comments.

Comment 2: *This paper that describes a clever way to make “water” membranes for gas separations. The method used to make very thin water membranes is especially interesting. The authors may want to incorporate the Supplementary Figure 1 into the main article, I believe readers will be drawn to it.*

Our Response: We appreciate the suggestion and considered including the fabrication figure in the main text. However, our goal in the main text is to emphasize water as the selective layer rather than the details of the fabrication method, which we view as supporting information for this proof-of-concept study. For this reason, we believe the figure is most appropriately placed in the Supplementary Information, while the main text is focused on the core concept. In a previous version of the manuscript, we included the fabrication figure in the main text, but we found it was distracting to readers.

Comment 3: *Stability of the water phase is a major concern, as the authors discuss. Pressure stability is one challenge, for which the most straightforward solution is to keep operating pressures well below the breakthrough pressure. How likely is that the feed contains contaminants that can change the surface tension of the water?*

Our Response: The reviewer raises an important point regarding stability under pressure. Stability of the water layer is governed by its surface tension, and contaminants that dissolve into the water can reduce that surface tension and lower the breakthrough pressure. In practice, most inorganic contaminants such as ammonia, H₂S, or SO₂ cause modest reductions in surface tension unless present at very high concentrations. Organic contaminants that strongly reduce surface tension such as surfactants are not expected in flue gas, biogas, or syngas streams that

motivate this work. Typical organic contaminants found in these gas streams consist of compounds such as short chain organics or volatile siloxanes. However, the partial pressures of these species are low enough that their dissolved concentrations in the water layer remain far below the levels needed to cause a significant decrease in surface tension. Figure 2e shows that even a large reduction in surface tension (~50%) would still allow for a breakthrough pressure above 30 bar for a 40 nm pore size membrane. This shows that the water layer can remain stable under realistic operating conditions, but operating well below the maximum displacement pressure is still recommended.

We have revised the manuscript as follows:

Line 430 – 440: “Moreover, because water is non-toxic and environmentally benign, this approach offers a sustainable and potentially regenerable alternative to solvent-based or polymeric membranes. Future efforts should prioritize scaling fabrication methods, evaluating long-term reliability, and assessing performance in full-scale process environments. *We demonstrate operation at pressures up to 72 bar without breakthrough, but condensation or dissolution of contaminants in the water layer (e.g., SO₂, H₂S, and light hydrocarbons) could reduce surface tension and lower that threshold. Although water is not prone to competitive sorption effects, future work should evaluate membrane performance under realistic multicomponent gas feeds and further assess long-term stability of the water layer, particularly under dry or low-humidity conditions.* With continued development, liquid water membranes may enable the high-performance gas separations essential for a sustainable economy.”

Comment 4: *The second stability issue is loss of the water phase, this may be the hard one.*

Our Response: We agree that loss of the liquid water phase is an important consideration. As noted in the manuscript, the Kelvin equation (Equation 5) shows that curvature of the liquid water meniscus in a hydrophilic pore lowers the equilibrium vapor pressure relative to bulk water. This reduction in equilibrium vapor pressure allows the water layer to remain stable at relative humidities below 100 percent and decreases the driving force for evaporation when the ambient water partial pressure is below the equilibrium value for the curved interface. These capillary effects help slow water loss and also provide additional stability for membranes with smaller pore sizes. We have added the following figure to the Supplementary Information:

Supplementary Fig. 15 | Reduction in equilibrium vapor pressure from the Kelvin effect for pore diameters less than 50 nm.

Comment 5: *Figure 1 i shows stability under pressurized conditions (why not plot actual permeance rather than relative permeance?), but the permeate side contains a droplet of water to avoid loss of water. Have experiments been carried out without that droplet? A requirement to keep the permeate side humidified is an obstacle for large scale implantation. The same has been proven true for facilitated transport membranes.*

Our Response: We thank the reviewer for this helpful comment. We chose to plot relative permeance rather than absolute permeance because the purpose of Figure 1i is to illustrate membrane stability rather than to highlight performance. The water droplet on the permeate side was part of the flow measurement setup, where changes in the height of the water column were used to determine gas flow rate. This setup also kept the permeate side saturated, which helped maintain the water layer. We note that, in practice, only a humidified feed would be required and that many real gas separation feeds are humid (e.g., biogas and flue gas). We have revised the text to address this:

Line 265 – 275: During the eight-day test, CO₂ permeance showed low variation, and CO₂:N₂ selectivity remained consistent between the first and eighth day of testing. This stability with dry feed conditions can be explained by the vapor pressure depression of water in nanoscale capillaries described by equation (5), with corresponding values given in Supplementary Table 4. By reducing the equilibrium vapor pressure of water in the pores, nanoscale confinement of water both increases the humidity threshold required for evaporation to occur and decreases the driving force when evaporation does take place. Under the tested conditions, these factors slow water loss from the dry feed side and promote net condensation from the permeate side, allowing the water layer to be sustained with only minimal moisture present. The humidity needed to maintain the water

layer is governed by the pore size dependent equilibrium vapor pressure described by the Kelvin equation (Supplementary Figure 15).

Comment 6: *In the stability test the feed side is close to being dry. What would happen if the feed contains meaningful amounts of water vapor? The water phase will “grow” and perhaps the full feed surface of the membrane (which is hydrophilic) will be coated with a film of water? This would reduce permeances.*

Our Response: We thank the reviewer for this question. We have observed that when the hydrophilic surface faces the feed side, exposure to humidified feed gas can cause the liquid water layer to grow. To avoid this, we oriented the membrane so that the hydrophilic side faced the permeate. With this orientation, any condensed water from a humid feed migrates into the permeate tubing instead of increasing the thickness of the water layer on the membrane surface. This approach allowed us to test feeds with meaningful humidity without causing growth of the water phase that would reduce permeance.

Comment 7: *In the Discussion section the stability issue is missing from the future efforts language. It deserves to be included here.*

Our Response: We appreciate the reviewer bringing this up. We have revised the manuscript as follows:

Line 430 – 440: “Moreover, because water is non-toxic and environmentally benign, this approach offers a sustainable and potentially regenerable alternative to solvent-based or polymeric membranes. Future efforts should prioritize scaling fabrication methods, evaluating long-term reliability, and assessing performance in full-scale process environments. We demonstrate operation at pressures up to 72 bar without breakthrough, but condensation or dissolution of contaminants in the water layer (e.g., SO₂, H₂S, and light hydrocarbons) could reduce surface tension and lower that threshold. Although water is not prone to competitive sorption effects, future work should evaluate membrane performance under realistic multicomponent gas feeds and further assess long-term stability of the water layer, particularly under dry or low-humidity conditions. With continued development, liquid water membranes may enable the high-performance gas separations essential for a sustainable economy.”

Reviewer #3:

Comment 1: *This manuscript presents a conceptually novel approach to gas separation using liquid water confined in sub-hundred-nanometer pores as the selective medium. The authors demonstrate that such membranes has high CO₂ permeance while maintaining high selectivity and can operate under high pressure and low humidity conditions. They further show that the*

separation mechanism follows a solubility-driven solution–diffusion model, where CO₂ selectivity arises from its much higher solubility in water relative to other gases. I think the concept is overall technically sound, elegantly simple, and quite novel. The mechanistic interpretation is coherent and supported by both theory and data. However, several aspects of the analysis and presentation require clarification and additional discussion, particularly regarding assumptions about pore geometry, active area normalization, and the completeness of the scale-up demonstration. That said, I'm in favor of the publication of this work after major revision.

Our Response: We thank the reviewer for their positive summary of our study and for recommending our work for publication. We have made major revisions to the manuscript to address their helpful comments, as described below.

Comment 2: *Reported permeance values are normalized assuming 12% membrane porosity. Given the tapered AAO pores, I am concerned that the true water filled area may deviate from this number significantly. The pore size distribution in an AAO membrane is usually not uniform. The largest pores will actually determine the onset of water being pushed out. I think the authors need to provide a quantitative pore-size histogram. Their pressure stability analysis may also need to be adapted accordingly*

Our Response: We appreciate the reviewer's comment regarding the pore structure and porosity of the AAO membranes used in our study. We have taken extensive electron imaging of these membranes (SEM and TEM) to verify the uniformity of the pore structure and porosity of the membranes. We have found that the pores are generally uniform and isotropic. ImageJ analysis of three SEM images confirms a surface porosity of $14.6 \pm 1.9\%$. TEM imaging shown in Supplementary Fig. 5 and included below confirms the uniformity of the AAO pores throughout the membrane thickness (i.e., we did not observe tapering of the pores). Water displacement measurements, which are skewed to the largest pores in the membrane, showed pore sizes that were consistent with those expected from imaging (Fig. 2e) We agree with the reviewer that a pore-size histogram would be beneficial to the manuscript. We have added the following figure to the Supplementary Information:

Supplementary Fig. 4 | (a) Binary SEM image of 40 nm pore size AAO membrane and (b) pore size distribution analyzed with ImageJ.

Supplementary Fig. 5 | Scanning transmission electron microscopy (STEM) high-angle annular dark-field image and element mapping images using energy dispersive spectroscopy (EDS). Images show the cross section of a 40 nm pore diameter AAO membrane sample sputtered with platinum. EDS elemental mapping is used to show the presence of platinum on the top surface of the AAO membrane as well as oxygen and aluminum within the membrane itself.

Corresponding changes to the main text are as follows:

Line 176 – 181: “The aluminum oxide membranes provided an ideal platform to explore the fundamental behavior and performance limits of water membranes because (i) they have uniform isopores of known pore size and porosity (Figure 2b,c and Supplementary Figure 4), (ii) they experience minimal deformation under pressure, allowing for reliable measurements, and (iii) they can be functionalized to trap water on the top surface using available surface science techniques (Figure 2d).”

Comment 3: *The scale-up experiments showcased the transferability of the concept but lack sufficient characterization. The authors should discuss the reasons for the lower permeance and pressure stability compared to the AAO tests. Even a qualitative discussion would strengthen this section substantially and may also provide guidelines for future works in this direction.*

Our Response: We agree with the reviewer that additional discussion on lower permeance and pressure stability of the commercial membranes would improve the manuscript. We have revised the text to include a quantitative discussion of the results and added a table to the SI. Changes to the main text are included below:

Line 391 – 408: “Testing with commercially available hydrophilic membranes showed that liquid water membranes demonstrate constant CO₂:N₂ selectivities around 40 regardless of the substrate used (Figure 5d), in agreement with membrane transport theory and our results with ceramic membranes. For the unoptimized microns-thick commercial membranes used in this work, the CO₂ permeance was relatively low (5.1 GPU for PVDF and 6.1 GPU for PES). We attribute this low permeance to the large thickness of the water layer, which likely filled the entire thickness of the hydrophilic membrane and resulted in a long diffusion pathway for gases. The thickness of the water layer in the commercial membranes was approximately 100 μm (Supplementary Table 6), compared to 190 nm for the fabricated membranes. The displacement pressure of water was 1.5 bar for PVDF and 6.6 bar for PES, pressures sufficiently high for low-pressure gas separation membrane applications but lower than those observed with alumina membranes due to the larger, polydisperse pore sizes. The larger pores are vulnerable to gas breakthrough at lower pressures. Based on the wetting pressure, the largest pores in the PVDF and PES membranes are 960 nm and 218 nm, respectively, far larger than the 40 nm pores used in fabricated membranes (Supplementary Table 6). Overall, results with commercial membranes show that large-scale porous substrates can be used as liquid water membranes, but further optimization to decrease the water layer thickness and membrane pore size is necessary to enable improved permeance and pressure tolerance.”

Supplementary Table 6. Commercial membrane properties assuming a porosity of 60% and a tortuosity of 2.

Membrane	Measured Membrane Thickness (μm)	Calculated Water Layer Thickness (μm)	Displacement Pressure (bar)	Specified Pore Size from Supplier (nm)	Calculated Maximum Pore Diameter (nm)
PVDF	110	116	1.5	450	960
PES	140	97	6.6	30	218

Comment 4: *The model assumes negligible gas-phase resistance. I wonder how true this is when the membrane is only partially filled with water and the gas needs to go through long and small pores.*

Our Response: We thank the reviewer for raising this point. In our experiments, the membranes were oriented such that the thin water layer was on the permeate side, and the feed gas filled the dry portion of the 50 μm anodized alumina pores. To directly assess the gas-phase resistance, we performed additional measurements with the water layer removed. Under these conditions, we observed a gas permeance exceeding 7×10^5 GPU, confirming that the AAO support presents minimal resistance to gas transport compared to even the thinnest water layer in this study.

We have added the below figure to the Supplementary Information:

Supplementary Fig. 18 | N_2 flux and permeance measured across 50 μm thick AAO with 40 nm diameter pores with no water layer. Permeance is normalized to 12% porosity.

We have revised the text as follows:

Line 464 – 466: The pressure at which the water layer is displaced from the pores of the membrane was determined by the measurement of abrupt changes in gas flow, up to 7×10^5 GPU for dry AAO (Supplementary Figure 17), from the permeate side of the membrane under applied pressure.

Comment 5: *Industrial gas streams may contain contaminants (e.g., SO_x, hydrocarbons, etc), which can change water properties (even with a small amount). A discussion on the potential impacts and mitigation would enhance the paper’s practical relevance.*

Our Response: We appreciate the reviewer raising the issue of contaminants in the feed stream. The stability of the water layer is governed by its surface tension, which sets the displacement pressure via the Young-Laplace equation. Contaminants that dissolve into the water layer can reduce the surface tension and lower the displacement pressure. While certain organic contaminants, such as surfactants, can significantly reduce surface tension even at low concentrations, these contaminants are not present in industrial gas streams such as flue gas, biogas, or syngas. More common contaminants include inorganics (e.g., SO₂, H₂S, NH₃) or small organics (e.g., methanol, formic acid, siloxanes), which do not decrease surface tension as significantly as surfactants. As shown in Figure 2e, even a 50% reduction in surface tension would maintain a displacement pressure above 30 bar for a 40 nm pore size membrane. However, we agree that careful pretreatment and operation well below the maximum breakthrough pressure are important controls for ensuring long-term membrane stability in industrial applications. We have revised the text as follows:

Line 430 – 440: “Moreover, because water is non-toxic and environmentally benign, this approach offers a sustainable and potentially regenerable alternative to solvent-based or polymeric membranes. Future efforts should prioritize scaling fabrication methods, evaluating long-term reliability, and assessing performance in full-scale process environments. We demonstrate operation at pressures up to 72 bar without breakthrough, but condensation or dissolution of contaminants in the water layer (e.g., SO₂, H₂S, and light hydrocarbons) could reduce surface tension and lower that threshold. Although water is not prone to competitive sorption effects, future work should evaluate membrane performance under realistic multicomponent gas feeds and further assess long-term stability of the water layer, particularly under dry or low-humidity conditions. With continued development, liquid water membranes may enable the high-performance gas separations essential for a sustainable economy.”

Comment 6: *The manuscript demonstrates 8-day stability under <1 % RH feed, but it would be more informative to estimate the water evaporation rate and expected drying timescale based on*

Eq. (5) and diffusion of water vapor through the gas phase. Even a simple order-of-magnitude estimate would help quantify long-term operational stability.

Our Response: We thank the reviewer for this question on evaporation rate under different feed gas humidities. Because interfacial evaporation occurs much faster than vapor can diffuse away, the partial vapor pressure at the gas–liquid interface is effectively fixed by the Kelvin-equilibrium value. We have added Supplementary Figure 15 to show the change in equilibrium vapor pressure with pore diameter based on the Kelvin effect.

Supplementary Fig. 15 | Reduction in equilibrium vapor pressure from the Kelvin effect for pore diameters less than 50 nm.

In our system, a dead-end test cell with no convection, water vapor transport between each interface and the distant gas reservoir is driven by diffusion through approximately stagnant gas. On the feed side, a dry gas cylinder maintains 0% humidity approximately 2 m from the feed side interface, driving evaporation. On the permeate side, a bubble flow meter maintains 100% humidity approximately 1 m from the permeate side interface, driving water vapor flux towards the membrane. The net evaporation rate can be modeled as the sum of evaporation on the dry feed side and condensation on the humidified permeate. This model predicts stability for roughly 2.5 days with dry feed gas, and adding humidity increases evaporation time (Response Figure 1). Realistic operating conditions would include crossflow in the feed and permeate which would decrease the water vapor diffusion length scale by orders of magnitude. In future studies we plan to better characterize stability by implementing crossflow and controlling humidity close to the interfaces on both sides of the membrane.

Response Fig. 1 | Modeled evaporation time as a function of feed gas relative humidity in a 50 μm thick membrane with 40 nm diameter pores. We assume the feed gas is humidity-controlled 2 m from the feed side interface and the permeate is controlled to 100% relative humidity 1 m downstream of the permeate side interface.

Comment 7: *Minor comments: In Eq. (4), θ is defined as the intrinsic contact angle. In fact, the relevant parameter for gas displacing water should be receding contact angle. The authors should note that the static angle that they measured can only serve as an approximation of the receding angle.*

Our Response: We thank the reviewer for bringing up the distinction between intrinsic and receding contact angle measurements. While the receding angle is more directly related to the gas-liquid displacement described by the Young-Laplace equation, it is difficult to measure accurately due to its sensitivity to experimental factors such as surface roughness, withdrawal rate, and evaporation. To provide a consistent estimate of wettability, we used the intrinsic contact angle measured on a flat, hydrophilic-modified silicon wafer. This allowed for a practical approximation of breakthrough pressure. We have revised the text as follows:

Line 144 – 146: where γ is the surface tension of the liquid-gas interface, θ is the intrinsic water contact angle on the membrane surface **which serves as an approximation for the receding contact angle**, and r is the pore diameter.

Comment 8: *8. In Eq. (6), r_c was used to represent minimum radius of curvature while already*

appearing in Eq. (5) to represent general radius of curvature. The authors should use a different symbol in this case.

Our Response: We appreciate the reviewer bringing up the distinction between radius of curvature and minimum radius of curvature constrained by the material. We have revised the text as follows:

Line 159:
$$r_{c,min} = \frac{r}{\cos\theta}$$

Comment 9: 9. GPU should be written in full in the abstract for broader accessibility.

Our Response: We thank the reviewer for this suggestion. We have revised the abstract as follows:

Line 42 – 44: “Reducing this thickness below 200 nm yields CO₂ permeances up to 11,600 gas permeation units GPU, with CO₂:N₂, CO₂:CH₄, and CO₂:H₂ selectivities of 40, 26, and 31, respectively, surpassing the performance of state-of-the-art membranes.”

Comment 10: 10. Please provide the error range for reported membrane thickness

Our Response: We thank the reviewer for highlighting the importance of including uncertainty in our measurements. The reported thickness was calculated using Equation S1 in the Supplementary Information and confirmed with cross sectional STEM-EDS chemical mapping shown in Supplementary Figure 4 and 5. The error range associated with these measurements is now summarized in Supplementary Table 1 (included below).

Supplementary Table 1. Hydrophilic layer thickness estimation.

Pore diameter (nm)	Incident angle	Calculated thickness (nm)	Estimated thickness from STEM-EDS chemical mapping (nm)
80	65°	172	180 ± 12
40	80°	227	230 ± 17

Reviewer #4:

Comment 1: I co-reviewed this manuscript with one of the reviewers who provided the listed reports. This is part of the Nature Communications initiative to facilitate training in peer review and to provide appropriate recognition for Early Career Researchers who co-review manuscripts.

Our Response: We thank the reviewer for their time and contribution to the review process, and we appreciate the Nature Communications initiative to involve and recognize Early Career Researchers in peer review.

Reviewer #5:

Comment 1: *This study is well done and presents an innovative new concept that is worthy of consideration and further development. It presents a simple, nature-inspired idea: use a thin layer of water held inside tiny, hydrophilic pores as a membrane for separating gases based on their differential solubility in water. The authors show very fast CO₂ transport with solid selectivity over common gases (N₂, CH₄, H₂), good stability under high pressure, a week-long continuous run, and an initial path to scale using off-the-shelf porous supports. The idea of using water in a supported liquid membrane format is interesting and can potentially be a disruptive approach to CO₂ separations. However, there are a few items that we would like the authors to address so that the data presented is in context and easy to follow and so that the readers can see the pros and cons of the approach and this study.*

Our Response: We thank the reviewer for their positive review. We have made changes in response to their comments to further strengthen the quality of our manuscript. Please see our point-by-point response to comments below.

Comment 2: *1. The biggest challenge I have is with the equivalency that the authors are making with plant systems in figure 1 and in the intro parts. This is a little bit of a stretch, while the stomata are indeed water filled they are hardly nanosized (more like micron- sized) and there is no concept of nanoconfinement there. If the authors want to then delve into nanoscopic confinement of water and gases dissolving in it then we approach the controversial topic of whether membrane proteins transport gas. See for example <https://doi.org/10.1002/cphc.201100034> . I think this controversy is still not resolved in the field. Further even if we talk about membrane proteins and transport through them, the mechanism is not strictly solution diffusion perhaps because there is single file diffusion in many such channels and this makes it hard to equate bulk phenomena like solution diffusion to it as discussed. I would urge the authors to soften this comparison as it is not really needed for the nice set of results they already have.*

Our Response: We thank the reviewer for this thoughtful comment and agree that the comparison to plant systems needs to be framed properly. The topic of plant physiology has been well-discussed in our group.

To clarify the biological context, stomata and substomatal cavities are air-filled structures and operate at the micron scale, as the reviewer notes. The water-filled channels in plant leaves are in the walls of mesophyll cells, which have pores approximately ten nanometers in diameter (Wang

et al., 2020, *Science Advances*, Ref 28). Evaporation and capillary forces in these water-filled channels generate the negative pressure (i.e., leaf water potential) needed to drive water up the xylem (Nobel, *Physicochemical and Environmental Plant Physiology*, 2020). The water-filled channels are also the primary location where CO₂ is dissolved and transferred to the photosynthetic pathways (Vesala et al., *Frontiers in Plant Science*, Ref. 26).

The principles of CO₂ uptake in plants inspired this work for two reasons. First, the water-filled nanochannels in plants are used to generate high negative pressure differences; these same pressure differences across air-liquid interfaces in hydrophilic capillaries are used in the membranes we demonstrate. Second, plants leverage water in the mesophyll cells as the medium to dissolve CO₂; similarly, our work also uses the uniquely high solubility of CO₂ in water as compared to other gases in its selective mechanisms.

We have revised the text to clarify the biological inspiration for this work:

Line 80 – 85: “Despite the promise of water as a medium for gas separations, relatively few studies have explored water in an engineered gas separation membrane. ~~5~~ Prior work has incorporated carbonic anhydrase, ionic liquids, or other reactive chemical into an aqueous layer to enable facilitated transport^{19,20,21}, ~~and none have demonstrated its unique advantages in CO₂ transport and pressure tolerance~~ but membranes that mimic the high-pressure tolerance of capillary-stabilized water observed in leaves to improve CO₂ separations remain unexplored.”

Comment 3: *Along the same lines, the title seems too general “Water as a gas separation membrane” while catchy is not strictly true. Water has to be confined or held within other membrane materials for the work to be possible so the authors could think about a more accurate title? I think the authors don’t emphasize enough that this concept is most analogous to supported liquid membrane, but they have turned the concept on its head and use water instead of solvents.*

Our Response: We thank the reviewer for this thoughtful comment. We carefully considered incorporating terms such as confinement or pores in the title but ultimately concluded that doing so may mislead readers regarding the mechanism we aim to emphasize. Although we operate with nanopores to stabilize the water and use nanoscale thicknesses, the phenomena that we observe are continuum in nature. Including “confinement” in the title risked implying that nanoscale confinement effects govern the separation mechanism, whereas several of our scaled-up experiments employ hydrophilic supports with conventional, micron-scale porosity.

We also discussed whether to reference supported liquid membranes (SLMs) directly in the title. However, because SLM terminology is field-specific, we felt that introducing it in the title might create confusion for non-specialist readers. That said, we fully agree that the conceptual link to SLMs is important. To do this, we have included sections in the manuscript to explicitly frame our system as a water-based analogue of a supported liquid membrane and have

added clarifying references throughout the Introduction and Discussion. Examples of these sections include:

Line 61 – 71: Supported liquid membranes, which consist of a porous support impregnated with a gas-selective liquid phase, have shown promise in addressing some limitations of current materials. The liquid used, commonly an ionic liquid or amine, can be tailored to selectively interact with CO₂ via favorable physical and chemical interactions, allowing for exceptionally high CO₂ selectivities. However, the gas permeance of supported liquid membranes is often limited by low gas diffusivity, difficulty in making thin liquid layers^{15,17}, and slow reaction kinetics in certain liquids that rely on chemical reactions with CO₂^{18,19}. In addition, many supported liquid membranes exhibit performance losses as CO₂ partial pressure increases due to saturation of chemical sorption sites, and experience liquid displacement or blowout at relatively low pressures (less than 5 bar)^{20,21}. Thus, there is a need to explore material systems that are more permeable, selective, and robust under pressure.

Line 192 – 194: The maximum measured displacement pressure is also at least three times higher than values for supported ionic liquid membranes in the literature, which range from 0.1–20 bar^{32,21}.

Line 252 – 257: Since transport resistances in the membrane are dominated by aqueous-phase diffusion in the liquid water layer, the observed CO₂ permeance of the water layer was inversely proportional to the membrane thickness, increasing from 46 to 11,600 GPU as the thickness decreased from 50 μm to 190 nm. In contrast, prior supported liquid membranes typically feature much thicker active layers in the order of tens of microns, constraining their achievable CO₂ permeances to below 1000 GPU^{17,34}.

Comment 4: 3. Lines 73-75 “Plants, in particular, have evolved to uptake CO₂ by dissolving it in water-filled nanochannels that line the walls of leaf mesophyll cells (Figure 1a)^{23–25}.” This according to my understanding is not correct but I am happy to look at any evidence the authors present.

Our Response: We thank the reviewer for this comment, which has also been a topic of discussion in our research group and with collaborators that study plant behavior. From our understanding, the general view of CO₂ uptake in the plant physiology community is that CO₂ enters the substomatal cavity in the plant through the open stomata. Once in the substomatal cavity, CO₂ partitions into water inside the wall of mesophyll cells. This process is described in the *Physicochemical and Environmental Plant Physiology* textbook authored by Nobel in Section 8.3 (Ref. 24) and also discussed in studies that examine the resistances involved in CO₂ uptake through mesophyll cells (doi: 10.1111/nph.16968). CO₂ uptake behavior has also been described as occurring in mesophyll capillaries in recent studies (Vesala et al., *Frontiers in Plant Science*, Ref. 26) and the nanopore terminology was also used in other cited papers (Wang et al., 2020, *Science Advances*, Ref. 28).

Comment 5: Line 211-212 also other places. “Gas flux was normalized to the membrane porosity of 12% so that reported values are representative of the gas transport rate through the water-filled active pore area.” I am not sure this is a fair comparison. The area should be the membrane area. Do other supported liquid membranes do this?

Our Response: We appreciate the reviewer raising this point about normalization. We chose to normalize gas flux by the membrane porosity to reflect the gas transport rate through the actual water-filled pore area because the focus in this work is on understanding the transport properties of water as the selective medium, rather than on optimizing the substrate geometry. While some studies on supported liquid membranes report fluxes normalized to total membrane area, others normalize to the effective pore area when characterizing the intrinsic performance of the liquid phase (Cheng et al., *Nature Communications*, 2025; Close et al., *Journal of Membrane Science*, 2012; Shahkaramipour et al., *Journal of Membrane Science*, 2014 ; Scovazzo et al., *Journal of Membrane Science*, 2004.) We have included the following table in the Supplementary Information and revised the text as follows:

Line 239 – 241: Experimentally measured gas permeances increased with aqueous gas solubility for N₂, H₂, CH₄, O₂, and CO₂ (Supplementary Table 2).

Supplementary Table 2. Gas permeance values as measured and normalized to the 12% membrane porosity.

	CO ₂ Permeance (GPU)	N ₂ Permeance (GPU)	O ₂ Permeance (GPU)	CH ₄ Permeance (GPU)	H ₂ Permeance (GPU)
Measured	1,390	34.9	65.1	66.1	44.3
Normalized to active area	11,600	291	543	551	369

Comment 6: Section “Permeance and selectivity performance in gas separations”. The discussion of selectivity and permeability in this paper and its comparison with conventional membranes is quite confusing and confused here. It seems like that the overall solution diffusion model is still valid here, the only advantage is that the thickness of the actual active part of the membrane can be minimized and is possible with this particular approach as opposed to traditional membranes or other supported liquid membranes. We think this section should describe this in a straightforward without convoluting arguments in favor of the current membrane.

Our Response: We thank the reviewer for pointing out the confusion in the discussion of the permeability-selectivity tradeoff and solution-diffusion transport membranes. The original

text went through a few iterations, and we agree that it was convoluted. We have revised the text as follows for clarity:

Line 278 – 284: “Conventional gas separation membranes are constrained by a trade-off where increases in gas throughput come at the expense of selectivity. In water-based membranes, a key advantage is the ability to reduce the thickness of the water layer without introducing defects, which are a major limitation in polymeric and supported liquid membranes. Since selectivity in the solution-diffusion framework depends only on differences in solubility and diffusivity between species (equation (3)), decreasing the water layer thickness enhances the gas permeance without affecting selectivity, allowing for operation with both high throughput and suitable discrimination between gases. ~~the parameters that govern selectivity and permeance are decoupled: selectivity is primarily determined by the relative solubility of gases in water, while permeance is dictated by the thickness of the water layer (equation (3)). Thus, decreasing the water layer thickness enhances the gas permeance without affecting selectivity, allowing for operation with both high throughput and suitable discrimination between gases.~~”

Comment 7: *We appreciate that the authors tried out conventional membrane platforms in addition to using anodized aluminum oxide membranes and have pointed out the challenges in scaling this technology based on the vast difference in performance seen.*

Our Response: We thank the reviewer for their positive comment and agree that further work on membrane design is required to effectively scale up this process.

Comment 8: *Line 310: There is an error in numbering Figure 4b and 4c*

Our Response: We thank the reviewer for pointing this out. Please see the corrected version below:

Line 315 – 317: “More importantly, these selectivities are sustained at CO₂ permeances exceeding 11,000 GPU, surpassing the performance of conventional membranes that typically operate in the range of 100–5,000 GPU (Figure 3b and 3c).”

Comment 9: *All data in the paper are on single gas experiments. While we understand the challenges of conducting mixed gas experiments on the current (really new) system, there should be some discussion on the current system’s expected performance with realistic mixed gas streams.*

Our Response: We thank the reviewer for acknowledging the difficulty behind mixed gas experiments. While we are currently unable to conduct mixed gas testing, we agree with the reviewer that a discussion on expected performance should be included in the manuscript. Please see the revised text below:

Line 430 – 440: “Moreover, because water is non-toxic and environmentally benign, this approach offers a sustainable and potentially regenerable alternative to solvent-based or polymeric membranes. Future efforts should prioritize scaling fabrication methods, evaluating long-term reliability, and assessing performance in full-scale process environments. We demonstrate operation at pressures up to 72 bar without breakthrough, but condensation or dissolution of contaminants in the water layer (e.g., SO₂, H₂S, and light hydrocarbons) could reduce surface tension and lower that threshold. Although water is not prone to competitive sorption effects, future work should evaluate membrane performance under realistic multicomponent gas feeds and further assess long-term stability of the water layer, particularly under dry or low-humidity conditions. With continued development, liquid water membranes may enable the high-performance gas separations essential for a sustainable economy.”

Response to Reviewers

We thank the five reviewers for their comments in both the first and second rounds of review. The feedback provided in the first round led to substantial improvements in the manuscript, and we have made additional revisions in response to the suggestions offered in this round. Below, we provide a point-by-point response to each comment and describe the revisions made in the manuscript. Reviewer comments are shown in black italic text, newly added material appears in blue, and deleted text is indicated with red strikethrough. All line numbers refer to the revised manuscript.

REVIEWER COMMENTS

Reviewer #1

Comment 1: *My overall impression is that the analogy to water uptake and transport in trees is over hyped. Water is transported in trees through combined transpiration (evaporation) and capillary stabilization in small pores like aquaporin channels. Here evaporation is to be avoided to maintain the water in the membrane pore channels.*

Our Response: We thank the reviewer for this comment and agree that the analogy to natural systems must be framed properly. Our intention introducing this reference was to highlight that water is frequently used as a medium for carbon dioxide uptake in nature. Plants in particular leverage two unique properties of water in their carbon dioxide uptake: (i) water has a high CO₂ solubility relative to other gases, and (ii) water has a high surface tension that allows for air-water interfaces to remain stable in nanoscale, hydrophilic pores, even under very high pressure differentials.

We agree with the reviewer that the analogy to water transport in trees should not be interpreted as a direct mechanistic comparison. In plants, water transport is driven by transpiration, whereas in our membrane system, evaporation is minimized to maintain water-filled pores. Our reference to plant systems was intended only to highlight capillary stabilization of liquid water under large pressure differentials and the use of water as a medium for CO₂ dissolution, rather than evaporation-driven transport.

We have revised the manuscript to clarify this distinction and avoid overinterpretation of the biological analogy.

Lines 74 – 87: “In nature, water plays a central role in gas separations: the two largest carbon sinks on Earth, oceans and plants, both rely on liquid water as a medium to absorb CO₂^{21,22}. Plants, in particular, have evolved to uptake CO₂ by dissolving it in water-filled nanochannels lining the walls of leaf mesophyll cells (Figure 1a)^{23–25}. **The gas–liquid interfaces in these channels** both serve as a platform to absorb CO₂ for photosynthesis and sustain the large negative pressures (up

to 150 bar) needed to drive water up from the roots using strong capillary forces^{26,27}. Nature therefore leverages two unique properties of water in its CO₂ uptake mechanism: (i) water has high CO₂ solubility through physical dissolution and (ii) water has a high surface tension that allows it to remain stable in nanoscale capillaries under high pressure differences²⁵. Despite these promising properties of water for gas separations, relatively few studies have explored water in an engineered gas separation membrane. Prior work has incorporated carbonic anhydrase, ionic liquids, or other reactive chemicals into an aqueous layer to enable facilitated transport^{3,28,29}, ~~and none have demonstrated its unique advantages in CO₂ transport and pressure tolerance~~ but membranes that mimic the high-pressure tolerance of capillary-stabilized water observed in leaves to improve CO₂ separations remain unexplored.”

Comment 2: *The novelty is limited in that there are review articles concerning supported liquid membranes for gas separation, e.g., ref.35, where it is stated, “capillary forces and are primarily responsible for solvent persistence; therefore, they contributes significantly to SLM stability by preventing the solvent from being escaped especially under high differential gas pressures.129–134 The capillary forces and hence the SLMs stability can be significantly increased by immobilizing the solvent in micropores or nanopores rather than mesopores or macropores, which prevents the solvent from escaping.*

Our Response: We thank the reviewer for this comment and agree that capillary stabilization of liquid phases in porous supports is well established in the supported liquid membrane literature, including the cited review. Our work does not claim novelty in the use of capillary forces to stabilize selective liquids, but in how they are applied and demonstrated in this system.

The key novel contribution of this study is that we demonstrate high permeance and pressure-tolerant membranes using ultrathin layers of liquid water. Specifically, this study demonstrates: (i) stable sub-micron water selective layers (down to 190 nm), which enable CO₂ permeances exceeding 11,000 GPU, an order of magnitude higher than typical SLMs that rely on 10–100 μm thick liquid layers; (ii) operation with pure liquid water as the selective phase, avoiding sorption saturation and reaction-rate limitations common to ionic liquid and facilitated-transport membranes, and maintaining permeability at CO₂ partial pressures up to 27 bar; and (iii) exceptional pressure tolerance, with experimentally measured water retention above 72 bar using nanoscale hydrophilic pores, exceeding reported stability ranges of most SLMs.

Examples of these discussions on these contributions are included below:

Line 61 – 71: “Supported liquid membranes, which consist of a porous support impregnated with a gas-selective liquid phase, have shown promise in addressing some limitations of current

materials. The liquid used, commonly an ionic liquid or amine, can be tailored to selectively interact with CO₂ via favorable physical and chemical interactions, allowing for exceptionally high CO₂ selectivities. However, the gas permeance of supported liquid membranes is often limited by low gas diffusivity, difficulty in making thin liquid layers^{15,17}, and slow reaction kinetics in certain liquids that rely on chemical reactions with CO₂^{18,19}. In addition, many supported liquid membranes exhibit performance losses as CO₂ partial pressure increases due to saturation of chemical sorption sites, and experience liquid displacement or blowout at relatively low pressures (less than 5 bar)^{20,21}. Thus, there is a need to explore material systems that are more permeable, selective, and robust under pressure.”

Line 192 – 194: “The maximum measured displacement pressure is also at least three times higher than values for supported ionic liquid membranes in the literature, which range from 0.1–20 bar^{32,21}.”

Line 252 – 257: “Since transport resistances in the membrane are dominated by aqueous-phase diffusion in the liquid water layer, the observed CO₂ permeance of the water layer was inversely proportional to the membrane thickness, increasing from 46 to 11,600 GPU as the thickness decreased from 50 μm to 190 nm. In contrast, prior supported liquid membranes typically feature much thicker active layers in the order of tens of microns, constraining their achievable CO₂ permeances to below 1000 GPU^{17,34}.”

Comment 3: *In Fig. 3, the authors selected only a limited set of membranes to which to compare their membranes and claim that they exceed the existing Robeson upper bound. Other supported liquid membranes like those described in ref. 28 along with carbon molecular sieves, e.g. Rahimalimamaghani et al. (Industrial & Engineering Chemistry Research 2023, 62, 45, 19116-19132) have CO₂/N₂ selectivities in excess of 150 and should be plotted in Fig. 3.*

Our Response: We thank the reviewer for this suggestion. In Figure 3, we selected CO₂/N₂ permeability-selectivity data from a range of membrane materials, including polymeric membranes, mixed matrix membranes, supported ionic liquid membranes, and metal organic framework containing membranes. These examples were chosen to provide a relevant benchmark of materials commonly discussed in relation to the Robeson upper bound.

In response to the reviewer’s comment, we have now added the high-selectivity CO₂/N₂ data reported by Rahimalimamaghani *et al.* to Figure 3a to broaden the comparison and include carbon molecular sieve membranes with high selectivity. The updated figure is shown below.

Comment 4: *I maintain that to be published in a top tier journal like Nature Comm, a membrane article should report mixed gas permeabilities. That could be done using simply GC analysis of the permeate and retentate streams compared to the feed stream.*

Our Response: We thank the reviewer for this comment and have completed mixed gas permeation tests to address this point. To do this, we collaborated with the National Lab of the Rockies to use their mixed gas filtration apparatus equipped with a GC as well as humidity, crossflow, and volumetric flow controls. The mixed-gas CO_2/N_2 selectivity was approximately 40 with a feed gas mixture of 90% N_2 and 10% CO_2 , consistent with the single gas measurements we had taken in our lab previously. We have revised the manuscript to incorporate this data as follows:

Lines 411 – 414 (Main text): “Multicomponent gas testing was conducted on a commercial PES membrane using a feed gas mixture of 90% N_2 and 10% CO_2 at a total pressure of 3.1 bar (Figure 5d). The mixed-gas CO_2/N_2 selectivity was approximately 40, consistent with single-component measurements, indicating minimal interaction between the two gases. Overall, results with commercial membranes show that large-scale porous substrates can be used as liquid water membranes, but further optimization to decrease the water layer thickness and membrane pore size is necessary to enable improved permeance and pressure tolerance.”

Line 492 – 496 (Methods): “Multicomponent gas testing under crossflow conditions was conducted using CO_2 and N_2 mass flow controllers on the feed side, with helium used as a sweep gas on the permeate side. Humidity was introduced to both the feed and permeate streams using water vaporizers and was maintained above 90% relative humidity, as measured by hygrometers installed on both sides of the membrane. Gas permeability and selectivity were determined by gas chromatography analysis of the permeate stream.”

Fig. 5 | Water membranes are scalable using commercial membrane substrates. **a**, Photograph of the hydrophilic polyvinylidene fluoride (PVDF) and polyethersulfone (PES) membrane samples. **b,c**, Scanning electron micrograph of the top surface of the hydrophilic PVDF and PES membranes, respectively. **d**, Measured CO₂ permeance and CO₂:N₂ selectivity of PVDF membranes using single-component gas feeds and PES membranes using both single-component and multicomponent gas feeds with trapped water layers. Error bars denote the mean ± 1 s.d. for measurements from at least three distinct membrane samples. **e**, Measured displacement pressures of water for PVDF and PES membranes.

Reviewer #2

Comment 1: *This reviewer (#2) was not part of (and did not see) the first round of reviews. The second round reviews by the other reviewers is quite extensive and covers topics not addressed by me. I have decided to stay out of those discussions.*

Our Response: We thank the reviewer for their time and for their comments during the previous round of review, which contributed to improving the quality of the manuscript.

Comment 2: *The author's response to my comment 6 raises a new question: All permeation experiments were carried out with the hydrophilic side facing the permeate? I must have missed it if there is any mention of this in the manuscript, but it certainly has to be disclosed. Furthermore, this membrane orientation works for single component permeation experiments but not for mixture experiments: concentration polarization in the pore section above the water phase will reduce selectivity and permeance. Something to be aware of when mixture experiments are carried out (which as other reviewers have mentioned is a required next step).*

It will take a minor revision to address my comment above, after which the manuscript can be published as far as this reviewer is concerned.

Our Response: We thank the reviewer for raising this point. In permeation experiments with the ultrathin water layer, the membranes were oriented with the hydrophilic water-filled side of the membrane facing the permeate, while the feed gas contacted the dry portion of the porous support. The membrane was oriented this direction to ensure that water drained out from the pores, and the selective layer was a consistent thickness. The membrane can also operate with the opposite orientation, but we wanted precise thickness control since we were reporting permeance as a function of the thickness in the manuscript.

We agree that future mixed gas testing should orient the water layer facing the feed to minimize concentration polarization. Based on our work so far, this orientation should operate without issue.

We have clarified the membrane orientation in the Methods section of the revised manuscript:

Line 486 – 487: “Fabricated thin membranes were oriented in the membrane cell with the thin, hydrophilic layer facing the permeate side to prevent condensation and accumulation of water on the feed side.”

Based on this comment, we estimated the impact concentration polarization would have, in theory, if the membrane were oriented as we tested it with mixed gas steams. Concentration polarization (CP) can be estimated by modeling the dry region of the pore as a 50 μm thick boundary layer:

$$\frac{C_{i,0}}{C_{i,b}} = \frac{\exp\left(\frac{J_{vf}\delta}{D_i}\right)}{1 + E_0 \left[\exp\left(\frac{J_{vf}\delta}{D_i}\right) - 1 \right]}$$

where $C_{i,0}$ and $C_{i,b}$ are the concentrations of CO₂ at the feed-side interface and in the bulk, respectively, J_{vf} is the velocity of transport through the membrane active layer (m/s), δ is the boundary layer thickness (50×10^{-6} m), D_i is gas diffusivity within the boundary layer, and E_0 is selectivity.

The most permeable membrane in this work had a permeance of 11,600 GPU with a pore diameter of 120 nm. With a pressure differential of 0.2 bar CO₂, similar to what was used in mixed-gas testing in the revised work, 11,600 GPU corresponds to a velocity of $J_{vf} = 0.0017$ m/s. The Knudsen diffusivity of CO₂ in a cylindrical pore with 120 nm diameter is estimated $D_i = 1.51 \times 10^{-5}$ m²/s. Using these values, we estimate a CP factor of 0.82. Increasing the pore size or decreasing the CO₂ partial pressure difference would increase the CP factor. For example, a CO₂ differential of 0.1 bar would result in a CP factor of 0.9.

We note that our testing with the ultrathin membranes was conducted with pure gas streams, so no concentration polarization occurred. However, even with mixed gases, the approximate CP factors are relatively low. As we suggest above, future work with multicomponent gas testing and crossflow should orient the water layer towards the feed side and control operating conditions to prevent water buildup on the active layer.

Reviewer #3

Comment: *The authors have addressed all my comments in the previous round. I now support the publication of this work in Nat. Comm.*

Our Response: We appreciate the reviewer's careful evaluation in the previous round of review and their time in assessing the revised manuscript. Their support for publication is gratefully acknowledged.

Reviewer #4

Comment: *I co-reviewed this manuscript with one of the reviewers who provided the listed reports. This is part of the Nature Communications initiative to facilitate training in peer review and to provide appropriate recognition for Early Career Researchers who co-review manuscripts.*

Our Response: We appreciate the reviewer's participation in the review process and their time spent evaluating the manuscript.

Reviewer #5

Comment: *I am satisfied with the revisions that the authors have made in response to reviewer comments.*

Our Response: We thank the reviewer for their comments during the previous round and their review of the revised manuscript. We are grateful for their assessment of the revisions.